# A Framework for Rapidly Predicting the Dynamics of Flexible Solar Arrays in the China Space Station with a Verification Based on On-Orbit Measurement Data

**Song Wu [1,2,3], Han Yan [2,3], Yuzhen Zhao [2,3], Yanhao Chen [2,3] and Guoan Tang [1,\*]**

1　Department of Aeronautics and Astronautics, Fudan University, Shanghai 200433, China; 20110290018@fudan.edu.cn
2　Shanghai Aerospace System Engineering Institute, Shanghai 201109, China
3　National Key Laboratory of Aerospace Mechanism, Shanghai 201109, China
\*　Correspondence: tangguoan@fudan.edu.cn

**Abstract:** The Chinese space station is a complex structure with large flexible appendages. Obtaining the on-orbit response characteristics of such a structure under different working conditions is a traditional and classic challenge in the field of dynamics. To address the on-orbit dynamics of the China Space Station, the basic equations for dynamic reduction, assembly and data recovery of linear and nonlinear substructures are derived based on the reduction and recovery theory, and a fast coupling analysis framework for flexible systems with nonlinear attachments is formed. This coupling analysis framework is adopted to quickly acquire the dynamic response of the China Space Station during in-orbit operation, thereby guiding the design. Taking SZ-15 radial docking to the Chinese Space Station as the object, the substructure of six nonlinear flexible arrays is reduced, the full flexible dynamic equation of the space station is assembled, and the response of each part of the flexible wing during the docking process is analyzed and recovered. By designing a reasonable and reliable flexible wing test scheme in-orbit, the acceleration at the root and top of the flexible wing during the docking of SZ-15 is obtained. The measured data in-orbit show that the acceleration analysis results of the typical parts of the flexible wing have a good agreement, which verifies the correctness of the fast coupling analysis framework of the flexible system. Hence, the dynamic coupling characteristics analysis of the main structure of the space station and the flexible wing based on this method can better guide the rationality of the design of the dynamic characteristics of the Chinese Space Station.

**Keywords:** flexible solar arrays; Chinese space station; a framework for rapidly predicting the dynamics; nonlinear substructure dynamic reduction; on-orbit measurement

## 1. Introduction

On 31 December 2022, China announced that the China Space Station had been fully completed and officially entered the conventional operation stage. The basic configuration of the China Space Station includes the Tianhe core module (hereinafter referred to as the core module) and two experimental modules, namely the Wentian Lab Module and Mengtian Lab Module, using a combination of the "T" configuration, as shown in Figure 1. There is a docking interface left at one end of the core module, which has the ability to further expand.

The space station is a large and complex system that has attracted widespread attention from international scholars. Anisimov et al. [1] studied the dynamic loading and structural strength analysis of the space station. Yang and Liu [2] investigated the vibration–attitude dynamics evolution and control of China's space station. Qiu et al. [3] presented a numerical simulation of a cabin ventilation subsystem in a space station oriented real-time system. In the numerous subsystems of the China Space Station, the flexible solar array is quite

significant because it provides over 90% of the energy. Its safety is a fundamental guarantee for the stable operation of the China space station. To provide sufficient energy for the space station, the area and length of the flexible solar array reach up to 100 square meters and 30 m, causing a significant vibration when the space station assembly undergoes orbit transformation or docking and separation with other spacecraft. The root and other parts of flexible solar array bear significant loads. It is needed to accurately predict the transient response, ensuring the flexible solar array has the required design safety factor. However, on one hand, the flexible solar array is subjected to pre-tension, with a fundamental frequency below 0.1 Hz and exhibits strong nonlinear characteristics. On the other hand, the space station is a vast and complex structure with significant flexibility features. The on-orbit load analysis for the flexible solar array of the space station needs to consider the flexibility of the space station structure and the nonlinear characteristics of flexible solar arrays. Therefore, the on-orbit load analysis for the flexible solar array of the China Space Station is a high-dimensional nonlinear transient solution problem with difficult convergence and slow iteration. How to quickly and accurately carry out the on-orbit load for the flexible solar array of the space station is a pressing problem during the engineering design stage.

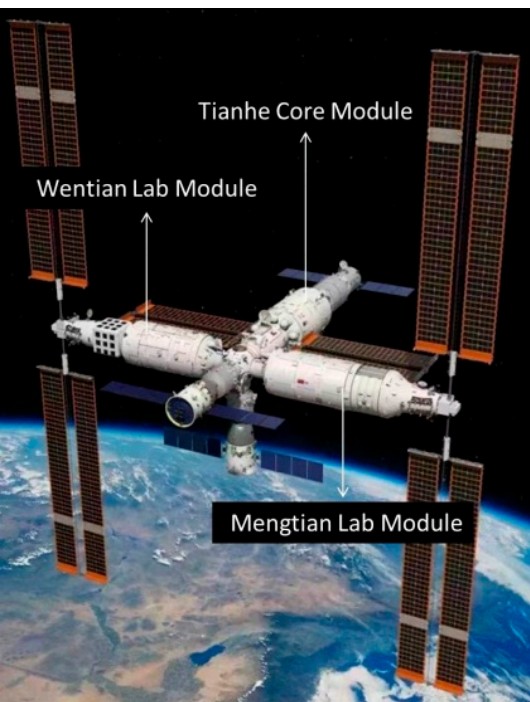

**Figure 1.** The basic configuration of the China Space Station.

For the on-orbit load analysis for complex aerospace assemblies with a nonlinear structure, Wu and Ghofranian [4] studied the dynamic issues of the flexible solar array and extension mechanisms of the International Space Station, summarized them and finally pointed out the difficulties in the research. The open-chain arm containing flexible links connected through telescopic joints is investigated so as to replace robotic chains with revolute joints, and it can be used in a fast and precise mechatronics system for utilization in space exploration, in a space station, and in spacecraft as well [5]. Aiming at the nonlinear vibration of a spacecraft with flexible joint attachments caused by joint nonlinearity, a set of explicit reduced order nonlinear ordinary differential equations for the motion of a flexible spacecraft with nonlinear joints is obtained, based on the global modal method [6]. An effective method is proposed for spacecraft designers planning to actively suppress the vibration of flexible solar array during the process of orbital maneuver [7]. In order to reduce the dimensionality of the dynamic equations, the constrained mode expansion

method was used to model the heavy-duty flexible manipulator and study the dynamic characteristics of the space station manipulator transformation system [8].

The China Space Station is a typical complex spacecraft completed through collaboration among multiple departments, for which the substructure method is usually utilized to analyze the dynamic characteristics [9,10]. As early as 1965, Hurty [11] first proposed the fixed interface modal synthesis method. In 1968, Craig and Bampton [12] improved Hurty's method to form the current fixed interface modal synthesis method, which is named the Craig–Bampton method. After decades of development, the substructure methods have become a significant analytical method in the field of structural dynamics, which are widely used in the aerospace [13–16]. For nonlinear problems in aircraft structures, a number of scholars have developed novel substructure methods from different perspectives in recent years. Kuether et al. [17] presented a nonlinear modal substructure method by using an additional polynomial function in the modal equations to characterize the geometric nonlinearity, and introduced a series of static loads to determine the nonlinear stiffness coefficients. Using this method, the geometrically nonlinear models can be built directly using commercial finite-element software such as Nastran-2012. Joannin et al. [18] extended the classic component mode synthesis methods by using the nonlinear complex modes of each substructure which were computed by means of a modified harmonic balance method. Through this method, the steady-state forced response of nonlinear and dissipative structures can be solved. Zhou et al. [19] presented a new impulse response function coupling analysis method by incorporating the hybrid rigid and nonlinear-elastic joints, which can take into account purely rigid, purely nonlinear-elastic or even hybrid rigid and nonlinear-elastic connections between substructures. Yuan et al. [20] developed an adaptive reduction approach to improve the component model synthesis-based reduction methods in the application of the assembled structure with frictional interfaces, which can significantly improve computational efficiency.

In this paper, for the analysis of in-orbit coupling dynamics of large complex spatial combinations containing multiple linear and nonlinear flexible structures, a data recovery method based on the reduced model is adopted to consider the cabin modules as a flexible body. On the basis of considering differential stiffness, linear and nonlinear flexible arrays are subjected to dynamic reduction and nonlinear flexible arrays are linearized to form a linear dynamics analysis model of a fully flexible system assembly. After completing the calculation, the data of each reduced model are restored and visualized, achieving the goal of considering the influence of nonlinearity and reducing the degree of freedom of analysis, effectively improving computational efficiency and adapting to the requirements of rapid iterative analysis in space station engineering.

## 2. Coupling Dynamics of Complex Spacecraft Assembly with Nonlinear Flexible Accessories

Complex spacecraft assembly generally consists of two or more modules, and each module carries several flexible accessories. For the coupling dynamic analysis of complex flexible spacecraft, a dynamic reduction method with fixed interfaces is generally used. The flexible accessories are considered as substructures and the structure of modules is considered as the residual structure. Although the dynamic reduction method can improve computational efficiency, it also has shortcomings. On one hand, it is mainly suitable for a linear structure. On the other hand, it is difficult to extract the internal response of the substructure required in engineering design, such as element force and stress. In fact, the substructure method for linear structures is very mature. However, there are few literature reports on nonlinear condensation methods that can be used in engineering practice for complex nonlinear structures represented by flexible solar arrays installed on the China Space Station. Therefore, in order to solve the on-orbit coupling dynamics problem of nonlinear flexible accessories in complex flexible spacecraft assembly, it is necessary to find new methods or improve existing methods.

For nonlinear flexible accessories, if they can be transformed into linear ones after appropriate processing, conventional dynamic substructure methods can be used to solve coupled dynamic problems in complex models. At the same time, after linearizing the nonlinear flexible accessory, the required internal response of the substructure can be quickly restored by defining corresponding parameters in the modal space based on the calculation results of the substructure modal coordinates. Therefore, for the coupled dynamics problem of complex spacecraft assembly with nonlinear flexible accessories, the linear module structure is divided into residual structure, and the flexible accessories are sequentially divided into multiple linear or nonlinear substructures, with a focus on the dynamic reduction in nonlinear flexible substructures and the recovery of internal response data in the substructures.

### 2.1. Dynamic Reduction in Nonlinear Flexible Accessory

Nonlinear problems are classified into three broad categories: geometric nonlinearity, material nonlinearity, and contact. Among them, material nonlinearity is an inherent property of any engineering material. Contact generally occurs in mechanism rather than structural dynamics analysis. And geometric nonlinearity effects are prominent in two different aspects: one is geometric stiffening caused by initial displacements and stresses, and the other is follower forces due to a change in loads as a function of displacements. The objective of applying a pretension which is induced by the coil spring to the flexible solar array is to obtain the required structural stiffness. For the flexible solar array of the China Space Station, it primarily exhibits geometric nonlinearity under the action of preload. Hence, the undamped dynamic problem equation corresponding to preload structures is as follows:

$$M\ddot{x} + (K + K_D)x = F \tag{1}$$

where, $K$ is the stiffness matrix of structure, $K_D$ is the differential stiffness matrix including geometric nonlinearity resulting from the initial load, $M$ is the mass matrix, and $F$ is the external excitation.

Define the stiffness matrix of the flexible solar array under the initial preload

$$\overline{K} = K + K_D \tag{2}$$

Obviously, as long as the differential stiffness matrix $K_D$ under preload is known, the dynamic problem of the nonlinear accessory is basically consistent with a conventional linear structure, and then dynamic reduction can be performed. The differential stiffness matrix under preload can be obtained by a structural nonlinear statics analysis method.

For any substructure, its degrees of freedom are composed of interface degrees of freedom set $x_j$ and non-interface degrees of freedom set $x_b$. And the substructure system dynamics equation is as follows:

$$\begin{bmatrix} m_{bb} & m_{bj} \\ m_{jb} & m_{jj} \end{bmatrix} \begin{Bmatrix} \ddot{x}_b \\ \ddot{x}_j \end{Bmatrix} + \begin{bmatrix} k_{bb} & k_{bj} \\ k_{jb} & k_{jj} \end{bmatrix} \begin{Bmatrix} x_b \\ x_j \end{Bmatrix} = \begin{bmatrix} f_b \\ f_j \end{bmatrix} \tag{3}$$

In the formula, $m$ is the mass matrix of the substructure, $k$ is the stiffness matrix of the substructure, and $x$ is the physical coordinates of each degree of freedom of the substructure; the subscripts refer to the block parts of each matrix.

There are three methods for substructure reduction: fixed interface, free interface, and mixed interface. Among them, the fixed interface method is most widely used in engineering due to its advantages of clear interface, high calculation accuracy, and stability. In the modal space of the substructure with a fixed interface, defining the following matrix

$$\Phi = \begin{bmatrix} \Phi_c & \Phi_s \\ 0 & I \end{bmatrix} \tag{4}$$

$\Phi_c$ is the main modes of substructure with a fixed interface, and the dimension is $b \times n$; $\Phi_s$ is the constraint modes of the substructure, and the dimension is $b \times j$, and can be expressed as $\Phi_s = -k_{bb}^{-1}k_{bj}$ according to its definition; $b$ is the non-interface degrees of freedom of the substructure; $n$ is the number of truncated modes; $j$ is the number of interface degrees of freedom of substructure.

Then, the coordinate transformation between the physical coordinates of nodes and modal coordinates is as follows:

$$\begin{bmatrix} x_b \\ x_j \end{bmatrix} = \begin{bmatrix} \Phi_c & \Phi_s \\ 0 & I \end{bmatrix} \begin{bmatrix} p_n \\ q_j \end{bmatrix} \tag{5}$$

Above, $p_n$ is the generalized coordinate corresponding to the main modes; $q_j$ is the generalized coordinate corresponding to the constrained modes, and is equal to $x_j$.

After substituting Equation (5) into Equation (3) and multiplying both sides of Equation (3) by $\Phi^T$ to the left, the substructure dynamics equation becomes as follows:

$$\begin{bmatrix} \overline{m}_{nn} & \overline{m}_{nj} \\ \overline{m}_{jn} & \overline{m}_{jj} \end{bmatrix} \begin{Bmatrix} \ddot{p}_n \\ \ddot{q}_j \end{Bmatrix} + \begin{bmatrix} \overline{k}_{nn} & \overline{k}_{nj} \\ \overline{k}_{jn} & \overline{k}_{jj} \end{bmatrix} \begin{Bmatrix} p_n \\ q_j \end{Bmatrix} = \begin{bmatrix} \overline{f}_n \\ \overline{f}_j \end{bmatrix} \tag{6}$$

Among them, the expressions for each block matrix are as follows:

$$\begin{aligned}
\overline{m}_{nn} &= \Phi_c^T m_{bb} \Phi_c; \\
\overline{m}_{nj} &= \Phi_c^T m_{bb} \Phi_b + \Phi_c^T m_{bj}; \\
\overline{m}_{jn} &= \Phi_s^T m_{bb} \Phi_c + m_{jb} \Phi_c; \\
\overline{m}_{jj} &= \Phi_s^T m_{bb} \Phi_s + m_{jb} \Phi_s + \Phi_s^T m_{bj} + m_{jj}; \\
\overline{k}_{nn} &= \Phi_c^T k_{bb} \Phi_c; \\
\overline{k}_{nj} &= \Phi_c^T k_{bb} \Phi_s + \Phi_c^T k_{bj}; \\
\overline{k}_{jn} &= \Phi_s^T k_{bb} \Phi_c + k_{jb} \Phi_c; \\
\overline{k}_{jj} &= \Phi_s^T k_{bb} \Phi_s + k_{jb} \Phi_s + \Phi_s^T k_{bj} + k_{jj}; \\
\overline{f}_n &= \Phi_c^T f_b; \\
\overline{f}_j &= \Phi_s^T f_b + f_j;
\end{aligned} \tag{7}$$

Generally, the main modes of substructures are normalized with respect to mass. Based on this, the above block matrix can be simplified as the following:

$$\begin{aligned}
\overline{m}_{nn} &= I_{nn}; \\
\overline{k}_{nn} &= \Lambda_{nn}^2; \\
\overline{k}_{nj} &= \Phi_c^T k_{bb} k_{bb}^{-1} k_{bj} + \Phi_c^T k_{bj} = 0; \\
\overline{k}_{jn} &= k_{jb} k_{bb}^{-1} k_{bb} \Phi_c + k_{jb} \Phi_c = 0;
\end{aligned} \tag{8}$$

where, $\Lambda$ is the circular frequency corresponding to the main mode of the substructure with fixed interface.

After the above derivation, the dynamic equation of the substructure system becomes as follows:

$$\begin{bmatrix} I_{ii} & m_{ij} \\ m_{ji} & m_{jj} \end{bmatrix} \begin{Bmatrix} \ddot{p}_i \\ \ddot{q}_j \end{Bmatrix} + \begin{bmatrix} \Lambda_{ii}^2 & 0 \\ 0 & \overline{k}_{jj} \end{bmatrix} \begin{Bmatrix} p_i \\ q_j \end{Bmatrix} = \begin{bmatrix} f_i \\ f_j \end{bmatrix} \tag{9}$$

where, the subscript $i$ is the order of the main mode of substructure, and the subscript $j$ is the order of interface degree of freedom; $\overline{k}_{jj}$ is stiffness matrix associated with modal parameters including nonlinear effects; $f_i$ and $f_j$ are modal force.

From Equation (9), it can be seen that after dynamic reduction, the degree of freedom of each substructure is the sum of the interface and the modal truncation number. Generally, the interface is a single node, whose degree of freedom is 6, and the number of modal truncation is typically on the order of hundreds or thousands. Therefore, the nonlinear flexible accessory with preload, such as flexible solar arrays, deployable mesh antennas

and so on, can be simplified as linear substructures with hundreds or thousands of degrees of freedom.

### 2.2. The Coupled Dynamic Equation of the Space Station System

After the reduction in substructures, it needs to be assembled with the residual structure to form the dynamic equation of the whole system structure. Without losing generality, assuming the system is divided into a residual structure and a substructure, the system dynamic equation with non-reduction in flexible modules and accessories is as follows:

$$
\begin{bmatrix} m_{aa} & m_{ab} & 0 \\ m_{ja} & m_{jj}^a + m_{jj}^b & m_{jb} \\ 0 & m_{bj} & m_{bb} \end{bmatrix} \begin{Bmatrix} \ddot{x}_a \\ \ddot{x}_j \\ \ddot{x}_b \end{Bmatrix} + \begin{bmatrix} k_{aa} & k_{ab} & 0 \\ k_{ja} & k_{jj}^a + k_{jj}^b & k_{jb} \\ 0 & k_{bj} & k_{bb} \end{bmatrix} \begin{Bmatrix} x_a \\ x_j \\ x_b \end{Bmatrix} = \begin{Bmatrix} f_a \\ f_j \\ f_b \end{Bmatrix} \tag{10}
$$

The subscript *a*, *b*, *j*, respectively, represents the residual structure (except the interface degrees of freedom), substructures (except the interface degrees of freedom) and the interface degrees of freedom; *m*, *k* and *f* are the mass matrix, stiffness matrix and the applied load vector, respectively.

For the fixed interface substructure reduction, by incorporating the substructure dynamics Equation (9) into the original system dynamics Equation (10), the system dynamics equation becomes as follows:

$$
\begin{bmatrix} m_{aa} & m_{ab} & 0 \\ m_{ja} & m_{jj}^a + m_{jj} & m_{ij} \\ 0 & m_{ji} & I_{ii} \end{bmatrix} \begin{Bmatrix} \ddot{x}_a \\ \ddot{q}_j \\ \ddot{p}_i \end{Bmatrix} + \begin{bmatrix} k_{aa} & k_{ab} & 0 \\ k_{ja} & k_{jj}^a + k_{jj} & 0 \\ 0 & 0 & \Lambda_{ii}^2 \end{bmatrix} \begin{Bmatrix} x_a \\ q_j \\ p_i \end{Bmatrix} = \begin{Bmatrix} f_a \\ f_j \\ f_i \end{Bmatrix} \tag{11}
$$

From Equation (11), it can be seen that after substructure reduction, the system degrees of freedom are basically the sum of the set of residual structural degrees and the modal truncation number of the substructure. Then, by solving Equation (11), the required dynamic response in the residual structure can be basically obtained, but only the generalized modal coordinates $p_i$ can be obtained for the substructure. To meet the demand for obtaining and visualizing the internal response data of the substructures in engineering, it is necessary to adopt a certain method to restore the required internal response data of substructures.

### 2.3. Data Recovery for Substructure with Dynamic Reduction

There are two methods for restoring the internal response of substructures. One is based on elastic mechanics, and the other is based on modal space. The substructure data recovery method based on modal space has a high computational efficiency and significant advantages for processing large models, which is very suitable for specific engineering problems.

According to the calculated generalized modal coordinates $p_i$ in the modal space, the internal node displacement response can be calculated as follows:

$$
x_b = \Phi_c p_i + \Phi_s q_j \tag{12}
$$

Here, $\Phi_c$ is the main modal under substructure fixed interface, whose number of columns is consistent with the order of modal truncation; $\Phi_s$ is the constraint modal of the substructure, and the order is consistent with the interface degrees of freedom.

For any substructure, the modal parameters in the modal space are defined, and the element modal strain can be given as follows:

$$
\Phi_\varepsilon = K_\varepsilon \begin{bmatrix} \Phi_c & \Phi_s \end{bmatrix}_e \tag{13}
$$

Above, $K_\varepsilon$ is the element geometry equation, and subscript *e* is the row corresponding to the element node degrees of freedom in modal space.

The element modal stress is as follows:

$$\Phi_\sigma = K_\sigma K_\varepsilon \begin{bmatrix} \Phi_c & \Phi_s \end{bmatrix}_e \tag{14}$$

Above, $K_\sigma$ is the element physical constitutive equation.
The element modal force is expressed as follows:

$$\Phi_f = K_e \begin{bmatrix} \Phi_c & \Phi_s \end{bmatrix}_e \tag{15}$$

Above, $K_e$ is the element stiffness matrix.

Based on the modal parameters defined above, the dynamic response of the internal element strain, stress and force of the substructure can be obtained in a similar way as shown by Equation (12). Similarly, other parameters in the modal space can be defined to obtain other types of response information within the substructure.

## 3. Coupling Dynamics Analysis of China Space Station during On-Orbit Docking

### 3.1. Configuration of the Space Station

Before the completion of construction, manned spacecraft and cargo spacecraft return regularly and dock with the core module. In the process of space station construction, it has a variety of combination configurations including line, L and T shape, as illustrated in Figure 2. After the construction of the China Space Station, it will be transferred to regular operation, and form a larger and more complex combination configuration.

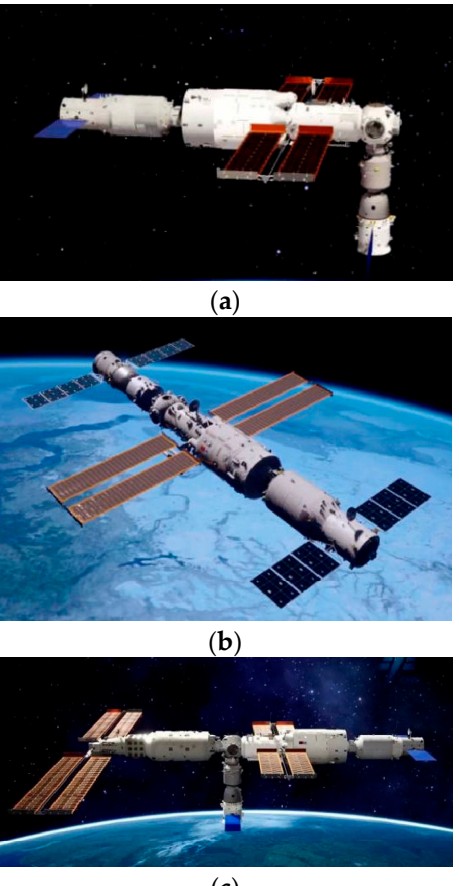

(**a**)

(**b**)

(**c**)

**Figure 2.** Typical configurations during the construction process of China Space Station. (**a**) L shape; (**b**) line shape; (**c**) T shape.

The docking process is a frequent action during the stable operation of the China Space Station in-orbit. The solar arrays have high flexibility and a weak load-bearing capacity.

During the docking process, flexible solar arrays are the most vulnerable component to damage. It is necessary to analyze the magnitude of the load on each part of the flexible solar array during the on-orbit docking process, verifying that the flexible solar array has the capacity to meet the on-orbit task.

The successful docking of SZ-15 with the space station is of great significance, achieving the maximum tonnage of the space station with three modules and three spaceships, and the maximum six astronauts in-orbit at the same time. In order to ensure the design safety of the flexible wing during the docking of SZ-15, it is necessary to conduct a fully flexible coupling dynamic analysis during the design stage of on-orbit task, extracting the on-orbit loads of various parts of the flexible solar array, especially the root of the entire array and the extension mechanism. Figure 3 shows the configuration of the space station and the flexible solar arrays during the docking of SZ-15.

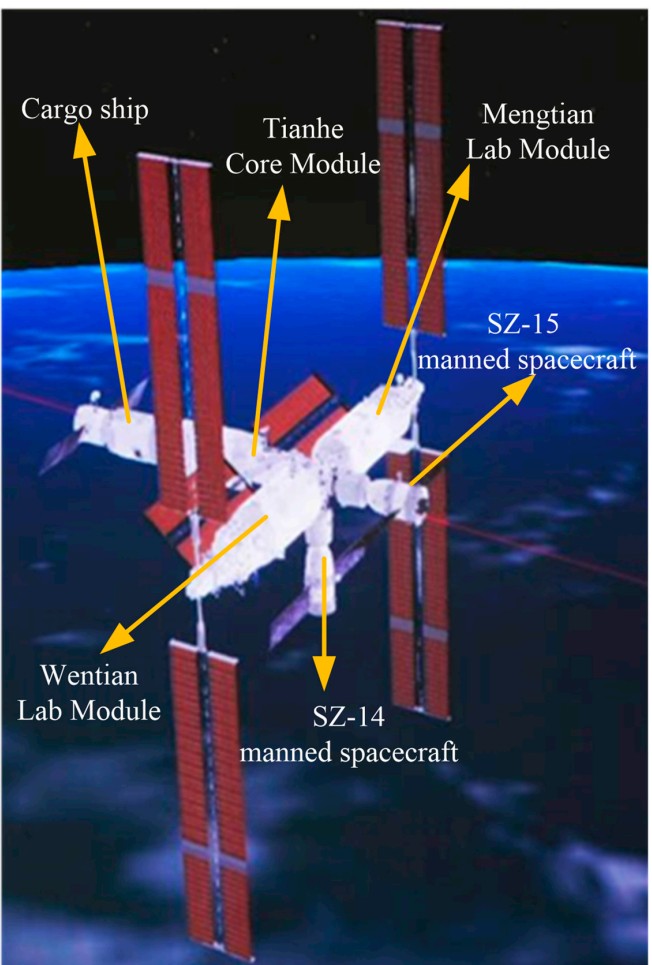

**Figure 3.** The configuration of China Space Station before SZ-15 docking.

*3.2. Modeling of the Space Station System*

During the orbital period of the China Space Station, the on-orbit dynamic response mainly exhibits the low-frequency vibration characteristics of the module structuraland the large flexible deployment attachment. The module structural loads are relatively small and negligible compared to the launch process.

The dynamic behaviors of the space station assembly are determined by the characteristics of the residual structure and flexible attachments. Thereby, the assembly can be divided into the residual structure and flexible accessories. The former includes cabins of Tianhe, Mengtian, Wentian and so on. And the latter refers to flexible solar arrays installed on these cabins.

### 3.2.1. Residual Structure

Based on the actual design status, a detailed dynamic analysis model for the space station assembly must be established, as shown in Figure 3. In order to adapt to the harsh dynamic environment during the launch stage of the rocket, the stiffness of a single module structure is often large. At the same time, we mainly focus on the dynamic characteristics of the space station assembly composed of multiple modules, so the modeling of a single module can be appropriately simplified.

Using Nastran-2012 divides the mesh of the modules and selects high-order precision elements to improve computational accuracy, such as 8-node quadrilateral elements and 6-node triangular elements. Furthermore, using two-dimensional shell elements to simulate thin-walled structure of the module, one-dimensional beam elements to simplify structural reinforcement, non-structural mass to simulate the mass of equipment, pipelines, propellants, etc., and using MPC or a bush element to equalize the connections between structures. As the docking is completed, the joint stiffness of the docking mechanism between the modules is large, and it can be equivalent to a bush element with a higher stiffness determined through ground static experiments.

However, before the docking is completed, the docking mechanism has a certain buffer effect on the impact of SZ-15, so the connection stiffness of the docking mechanism during the docking process is different from that after the docking. Through the semi-physical dock and buffer test of the docking mechanism on the ground, the stiffness of the docking mechanism during the docking process can be determined and shown in Table 1. Finally, the dynamic model of the residual structure is established using the MSC.NASTRAN, which is illustrated in Figure 4. The coordinate system in Figure 4 is the reference coordinate system in the subsequent analysis. If there is no special explanation, the direction of the mass inertia and other contents shall be referred to this coordinate system. The X direction is along the axis of the core module, and the positive point towards the side of the Wentian module; the Z direction is along the axis of the Mengtian and Wentian modules, and the positive point towards the side of Wentian module; the Y direction is determined by the right-hand rule.

**Table 1.** The equivalent stiffness of the docking mechanism during the docking process.

| Orientation | Magnitude | Unit |
|:---:|:---:|:---:|
| axial | $3.64 \times 10^6$ | N/m |
| transverse | $7.04 \times 10^8$ | |
| twist | $3.31 \times 10^7$ | Nm/rad |
| bend | $7.65 \times 10^6$ | |

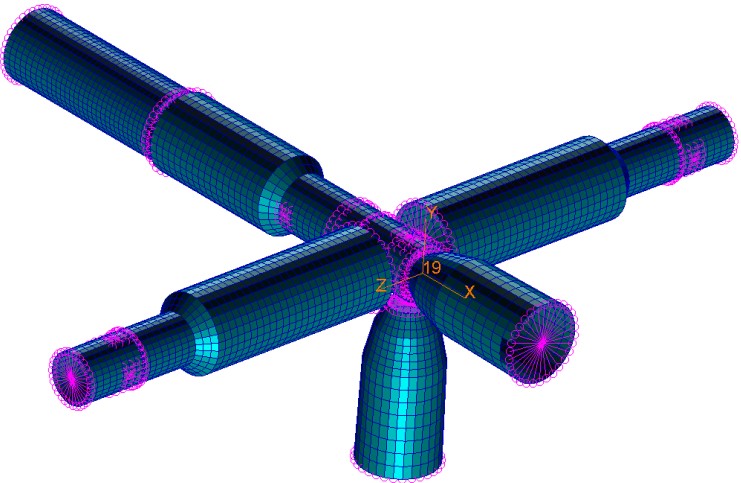

**Figure 4.** The dynamic model of China Space Station assembly.

The space station assembly is without constraint during in-orbit operation, so the dynamic characteristics of the residual structure without constraint should be analyzed. Based on the finite element model established in the Nastran-2012, the frequency and mode shape of the residual structure were obtained through modal analysis. The first 10 non-zero modal frequencies are shown in Table 2.

**Table 2.** The first 10 non-zero modal frequencies of the residual structure.

| Serial Number | Frequency (Unit: Hz) |
| --- | --- |
| 1. | 0.675 |
| 2. | 0.849 |
| 3. | 0.931 |
| 4. | 1.842 |
| 5. | 2.168 |
| 6. | 2.552 |
| 7. | 2.676 |
| 8. | 3.655 |
| 9. | 4.071 |
| 10. | 4.618 |

During the docking process, the low-frequency dynamic characteristics of the space station module structure are mainly presented, so the main focus is on the first three non-zero modes of the residual structure. The vibration mode cloud diagram is shown in Figure 5.

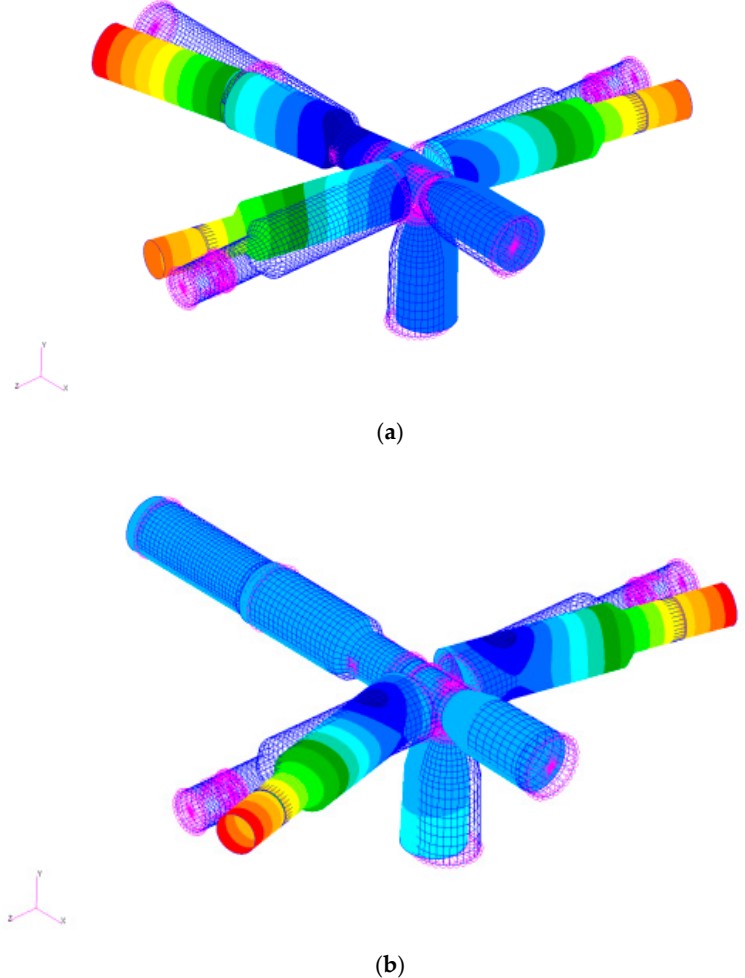

(**a**)

(**b**)

**Figure 5.** *Cont*.

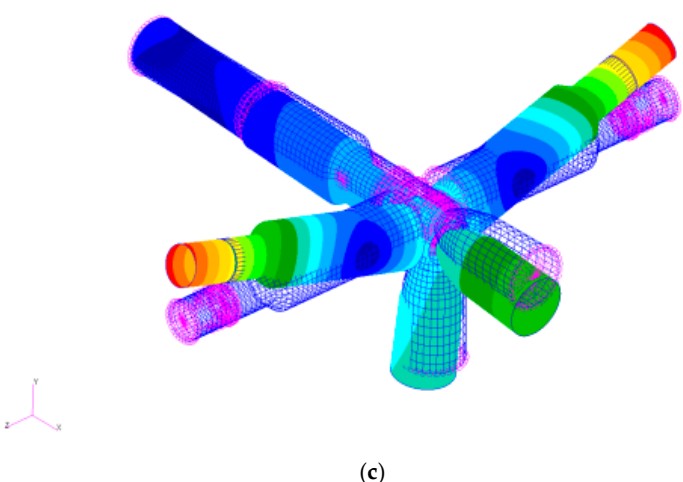

(**c**)

**Figure 5.** The vibration mode cloud diagram of the first three non-zero modes of the residual structure. (**a**) The first non-zero mode; (**b**) the second non-zero mode; (**c**) the third non-zero mode.

To facilitate the description of the vibration mode of the residual structure, ignoring the Y-direction manned spacecraft, the space station assembly can be viewed as a cross shaped configuration formed by the intersection of two lines. One line is composed of Mengtian and Wentian modules, and the other is composed of cargo spacecraft, the core module and the manned spaceship. The first non-zero modal vibration mode is characterized by the rotational vibration of the entire space station around the Y-axis, and the rotational phases of the two lines are opposite. The second non-zero mode vibration is manifested as the line where Mengtian and Wentian modules are located bending in the -Y direction, and the line where the core module is located translating in the Y direction. The third non-zero mode vibration is characterized by the bending of the line where Mengtian and Wentian modules are located in the -X direction, and the translation of the line where the core module is located in the X direction. From the docking position and direction of SZ-15 in Figure 3, it can be seen that the vibration of the space station assembly during the docking process is mainly manifested as the free vibration of the third non-zero mode, followed by the second order free vibration as the Y-axis asymmetry caused by radial manned spacecraft.

### 3.2.2. Flexible Solar Arrays

The space station has a total of six flexible solar arrays, with two carried by the core module, Wentian module, and Mengtian module, respectively. The two flexible solar arrays on the core module are basically the same, with a length of approximately 12 m; the four flexible solar arrays on the lab modules (Wentian module and Mengtian module are called the lab module) are basically the same, with a length of approximately 30 m. Therefore, for ease of description, the six solar arrays on the space station can be divided into two categories according to the length of the flexible wings, which are the flexible solar arrays of the core module and the flexible solar arrays of the lab module. Although there is a significant difference in length between the two flexible solar arrays of the core module and lab module, the structure and principle are basically the same, the tension applied to the array surface especially is also basically the same. Therefore, one of the flexible solar arrays can be taken as an example to introduce the dynamic modeling process.

The flexible solar array mainly consists of a lifting mechanism, a stretching mechanism, a deployable mechanism, a guiding mechanism, a driving mechanism, a storage box, a storage container, a battery array, a cable system, etc. Its state after fully deploying in-orbit is shown in Figure 6. The flexible solar array obtains the required stiffness through the tension applied by the tensioning mechanism and maintains a flat state. The single side array when fully deployed is shown in Figure 7.

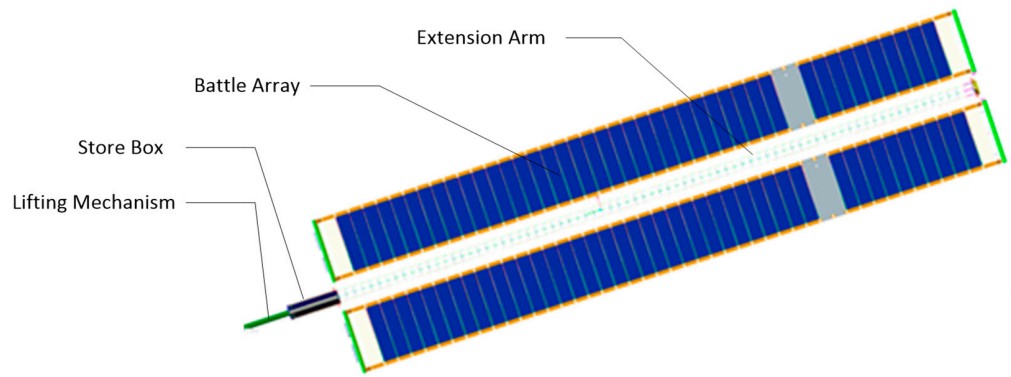

**Figure 6.** Schematic diagram of flexible solar array.

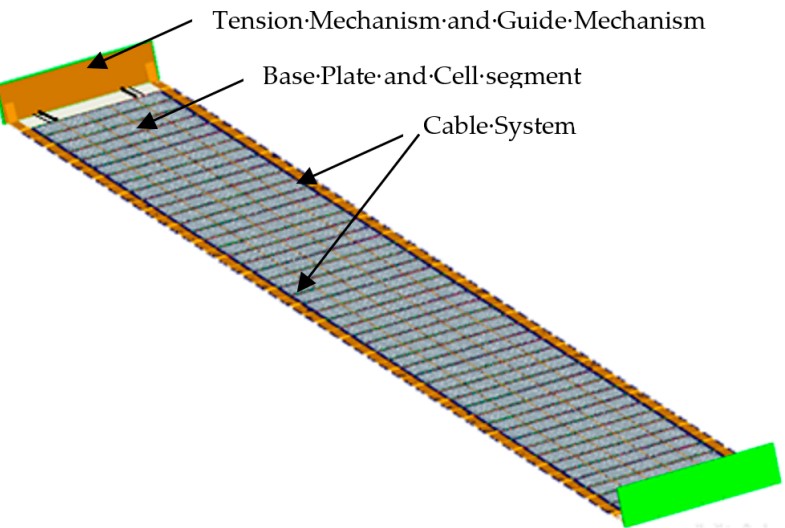

**Figure 7.** The single side array with fully deployed.

Based on the design and working principle of the flexible solar array, the beam and shell elements are mainly used to model the structure, and to simulate the connections between the structures via linear and nonlinear spring elements. The initial tension of the tensioning mechanism acting on the flexible solar array is simulated by applying element temperature stress or the gap element to achieve the stiffness of the flexible solar array under the tensioning force. Here, we focus more on the dynamic characteristics of the space station system, and more details on the modeling of flexible arrays can be found in the References [21,22]. Finally, the finite element model for the flexible solar array of the core module is established as shown in Figure 8.

Then, a nonlinear static analysis method is used to solve the dynamic characteristics of the flexible solar array under tension and root fixed support constraints, taking into account the differential stiffness of the preload and geometric nonlinear characteristics. To improve the accuracy of the analysis model, if ground modal tests cannot be conducted on the fully deployed flexible solar array, a component level or principal ground test should be carried out to correct the nonlinear dynamic model. Following this ground test correction, the initial deformation of the flexible solar array of the core module under initial tension using nonlinear analysis methods is shown in Figure 9.

Using the results from the non-linear analysis as the initial pre-stress for the modal analysis, the modes of the flexible solar array of the core module can be calculated according to the method introduced in Section 2.1; the main modal shapes and frequencies are shown in Figure 10 and Table 3.

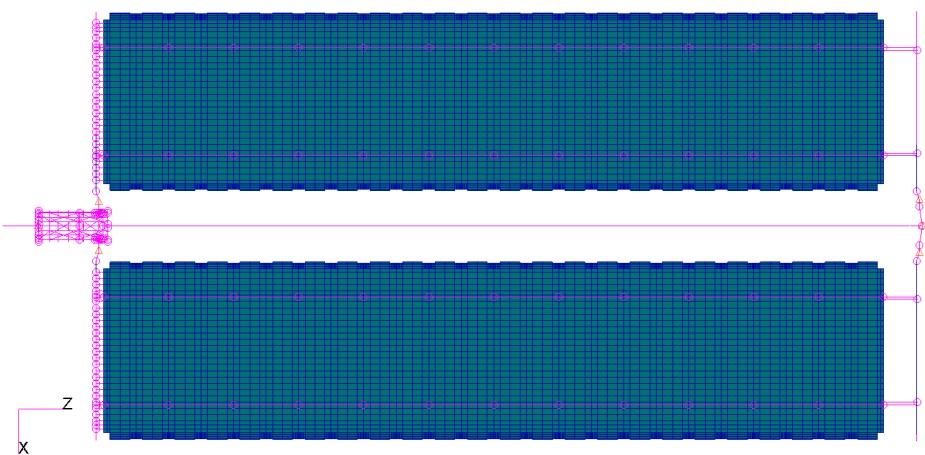

**Figure 8.** FEM model of flexible solar array of core module.

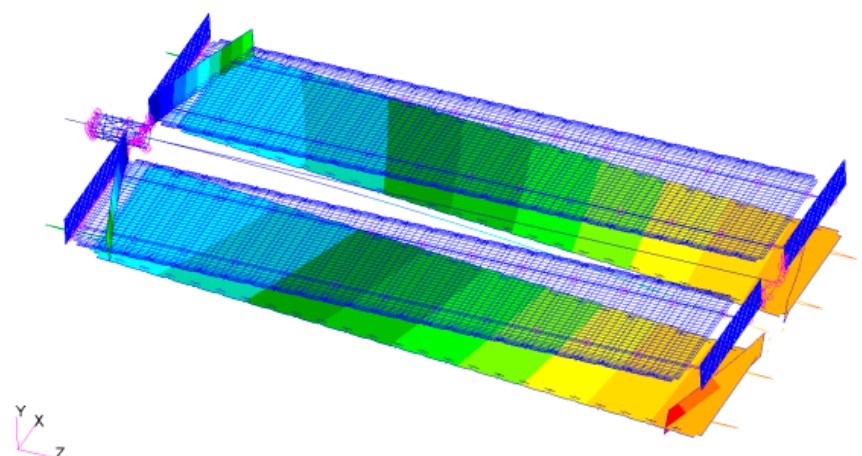

**Figure 9.** Schematic diagram of initial static deformation of flexible solar array of core module.

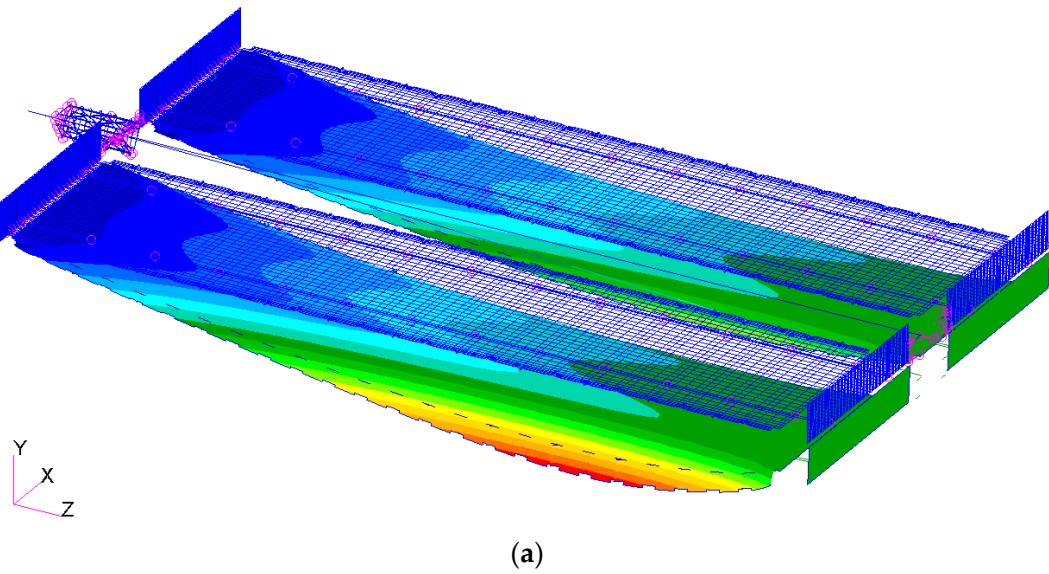

(**a**)

**Figure 10.** *Cont*.

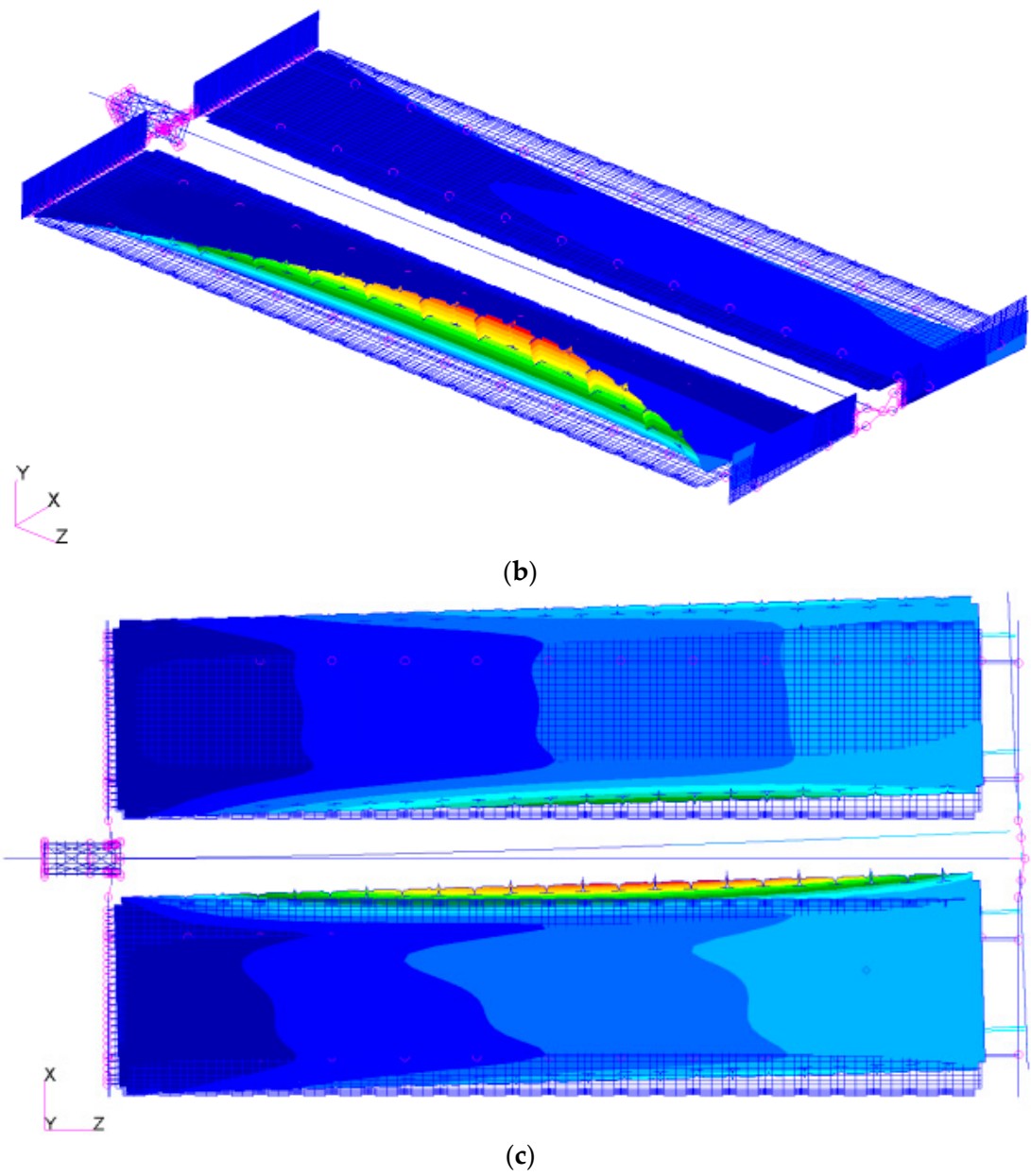

**Figure 10.** Main mode shapes and frequencies of flexible solar array of core module. (**a**) The 1st out-plane bending mode; (**b**) 1st torsional mode; (**c**) 1st in-plane bending mode.

**Table 3.** The main model frequency of flexible solar array of core module.

| Model Number | Frequency (Unit: Hz) | Model Shape Description |
| --- | --- | --- |
| 1 | 0.113 | 1st out-plane bend |
| 6 | 0.130 | 1st twist |
| 7 | 0.158 | 1st in-plane bend |

By using the same method and steps to establish the dynamic model of the flexible solar array of the lab module, we can carry out the nonlinear statics analysis and free mode analysis. The initial deformation of the flexible solar array of the lab module under initial tension and root fixed support constraints using nonlinear analysis methods is shown in Figure 11. The main modal shapes and frequencies are shown in Figure 12 and Table 4.

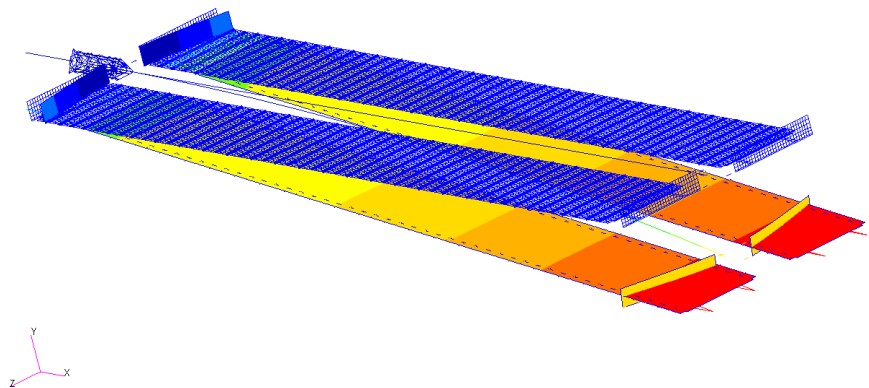

**Figure 11.** Schematic diagram of initial static deformation of flexible solar array of the lab module.

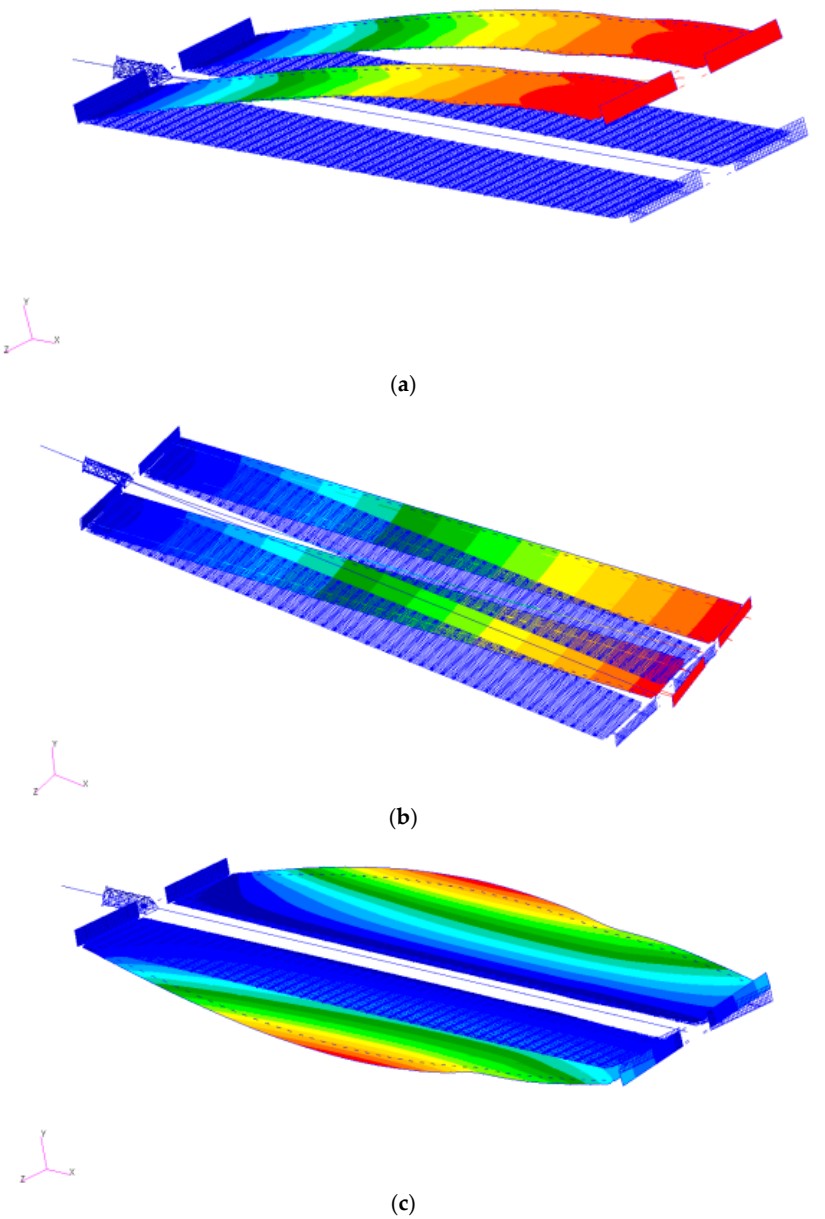

(**a**)

(**b**)

(**c**)

**Figure 12.** Main mode shapes and frequencies of flexible solar array of the lab module. (**a**) The 1st out-plane bending mode; (**b**) 1st in-plane bending mode; (**c**) 1st torsional mode.

**Table 4.** The main model frequency of flexible solar array of the lab module.

| Model Number | Frequency (Unit: Hz) | Model Shaped Description |
|:---:|:---:|:---:|
| 1 | 0.041 | 1st out-plane bend |
| 2 | 0.046 | 1st in-plane bend |
| 3 | 0.082 | 1st twist |

After completing the dynamic modeling of a single flexible solar array, in order to facilitate the connection between the substructure and the residual structure during dynamic reduction, the substructure and residual structure were assembled to form a fully flexible composite system dynamic model, as shown in Figure 13.

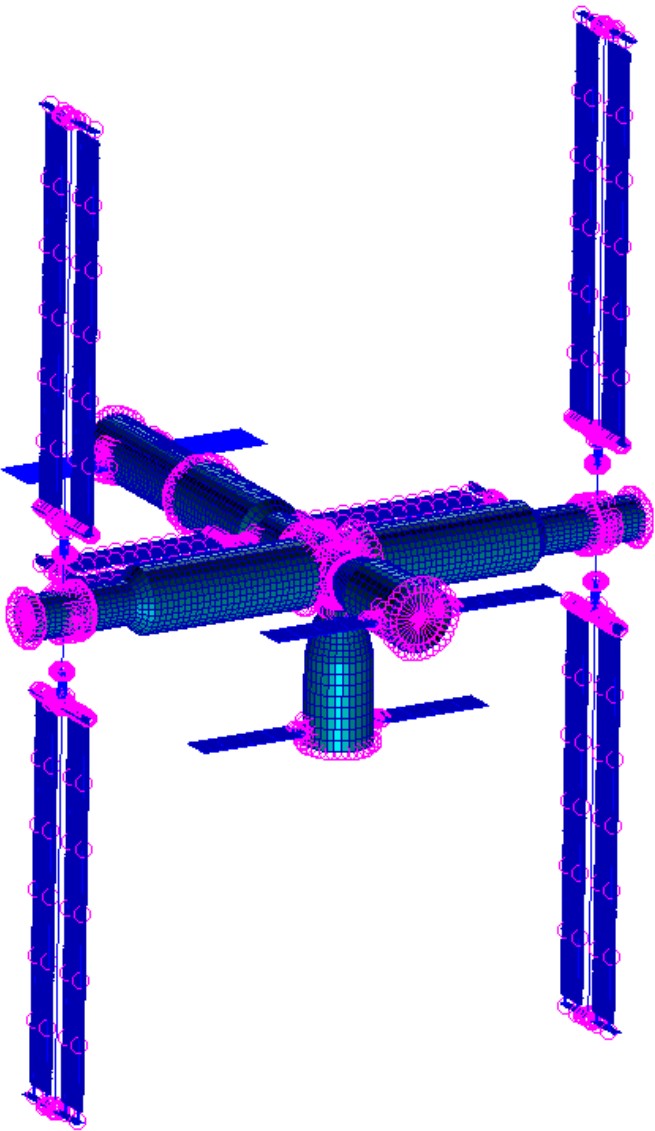

**Figure 13.** Fully flexible dynamic model of the space station assembly.

Due to the structural frequency of the space station assembly modules being less than 1 Hz, the flexible solar array retains modal information within 3 Hz during the dynamic reduction in the flexible solar array using the method introduced in Section 2. Among them, the flexible solar array of the core module retains about 322 orders, while the flexible solar array of the lab module retains about 665 orders.

### 3.3. Dynamics Characteristics of the Space Station System

The flexible solar array in-orbit is actually a free constrained boundary state under the space station system, so the frequency characteristics of the flexible solar array in-orbit may be very different from the fixed constrained boundary state, unless the mass inertia of the residual structure of the space station is much larger than that of the flexible solar array. Referring to the coordinate system shown in Figure 4, the mass characteristics of the residual structural are shown in Table 5, and the mass characteristics of the flexible solar array of the core module and lab module are shown in Tables 6 and 7, respectively.

**Table 5.** The mass characteristics of residual structural.

| Orientation | Inertia Tensor (Unit: kgm$^2$) | | | Mass (Unit: kg) |
|:---:|:---:|:---:|:---:|:---:|
| | **X** | **Y** | **Z** | |
| X | $5.9 \times 10^6$ | $2.5 \times 10^5$ | $2.1 \times 10^2$ | |
| Y | $2.5 \times 10^5$ | $1.2 \times 10^7$ | $4.8 \times 10^{-1}$ | 95,000 |
| Z | $2.1 \times 10^2$ | $4.8 \times 10^{-1}$ | $6.6 \times 10^6$ | |

**Table 6.** The mass characteristics of the flexible solar array of core module.

| Orientation | Inertia Tensor (Unit: kgm$^2$) | | | Mass (Unit: kg) |
|:---:|:---:|:---:|:---:|:---:|
| | **X** | **Y** | **Z** | |
| X | $7.8 \times 10^3$ | 0.43 | 0.22 | |
| Y | 0.43 | $8.6 \times 10^3$ | −0.14 | 340 |
| Z | 0.22 | −0.14 | $7.5 \times 10^2$ | |

**Table 7.** The mass characteristics of the flexible solar array of the lab module.

| Orientation | Inertia Tensor (Unit: kgm$^2$) | | | Mass (Unit: kg) |
|:---:|:---:|:---:|:---:|:---:|
| | **X** | **Y** | **Z** | |
| X | $5.3 \times 10^4$ | −4.8 | 0.28 | |
| Y | −4.8 | $1.3 \times 10^3$ | −0.71 | 590 |
| Z | 0.28 | −0.71 | $5.4 \times 10^4$ | |

The ratio of the mass inertia between the residual structure of the space station and the flexible solar array reaches more than 100 times. It can be concluded that the frequency of the flexible solar array under the constraint boundary of the fixed support is basically consistent with the on-orbit state. At the same time, the system dynamics characteristics of the residual structure and the flexible solar array of the space station are analyzed and verified. As described in Section 2.2, the mass and stiffness matrix of the residual structure and the flexible solar array reduction model are assembled to solve the dynamic characteristics of the flexible solar array substructure in the space station system. The frequency characteristics of the flexible solar array in two boundary states are shown in Table 8.

**Table 8.** Frequency characteristics of flexible solar array in two boundary states of root support and space station system (unit: Hz).

| Name | | The Root Fixed | Free in the Space Station System |
|:---:|:---:|:---:|:---:|
| Flexible solar array of core module | 1st out-plane bend | 0.113 | 0.116 |
| | 1st in-plane bend | 0.158 | 0.160 |
| Flexible solar array of the lab module | 1st out-plane bend | 0.041 | 0.043 |
| | 1st in-plane bend | 0.046 | 0.048 |

The results from Table 8 show that the frequency characteristics of the flexible solar array on-orbit state are basically consistent with the constraint boundary of the root fixed support, and this is the same as the prediction results of inertia difference.

### 3.4. Transient Dynamics Analysis of Docking Process

3.4.1. External Excitation

During the docking process of the manned spacecraft with the space station, the relative speed of the two spacecraft is controlled through on-orbit rendezvous and docking technology. After gradually approaching, the docking mechanism is used to connect the SZ-15 spacecraft and the space station. Due to the initial relative velocity during docking, the space station assembly generates a certain vibration response, which gradually decays to zero under the damping effect. Previous on-orbit docking shows that the initial docking speed is basically maintained at $(0.25 \pm 0.1)$ m/s. Therefore, a constant initial velocity was set for SZ-15 spacecraft during the analysis of the docking transient dynamics. The application is shown in Figure 14.

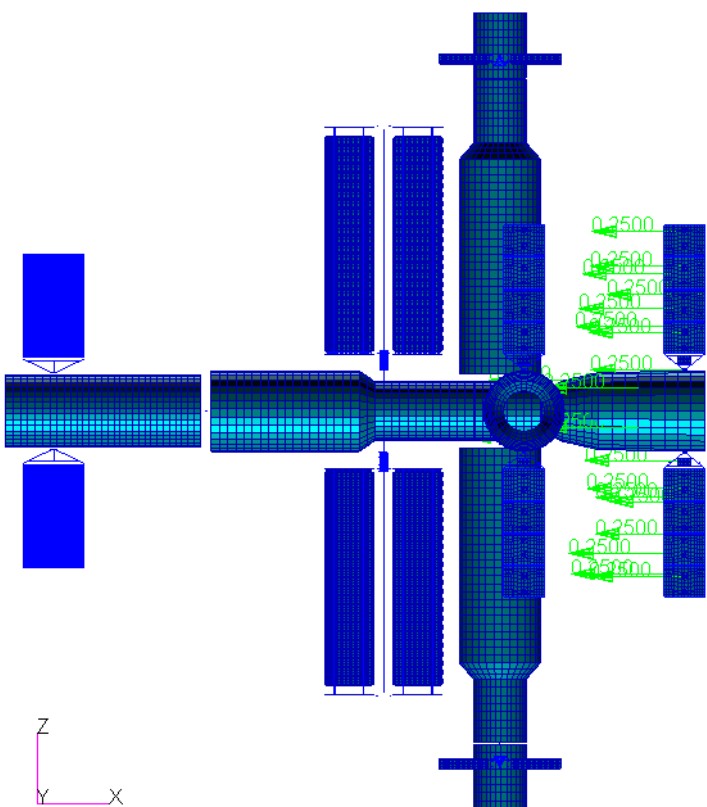

**Figure 14.** Initial speed during the docking of SZ-15 spacecraft.

3.4.2. Analysis Parameter

In the analysis, the free boundary is used to simulate the free state of the space station in-orbit, and the dynamic equations of the residual structure and each flexible solar array structure is assembled to form the space station system dynamic equation described in Section 2.2. The total degree of freedom of the system was not much after the reduction, so the direct method was used to solve the problem; the damping size of the structure was 0.05. The analysis time step was 0.01 s and the total calculation time was 40 s, and the residual structural response and the response of the generalized modal coordinates of the substructure were calculated. Finally, the response information of other parts of the substructure were obtained according to the method of data recovery within the substructure described in Section 2.3.

*3.5. Calculation Results*

The residual structure is an actual physical model, the degree of freedom is not reduced, and the required response information can be obtained more easily. Its calculation results are relatively conventional, and will not be explained separately here. For the substructure, the response information of any part of the substructure can be obtained by using the method presented in this paper, and the effect of obtaining and visualizing any response can be achieved by the uncondensed model. The analysis node results of the flexible solar arrays refer to the coordinate system in Figure 4.

### 3.5.1. Displacement

The internal displacement data of a single flexible solar array can be recovered and stored in a separate file, and the results of the displacement response can be visualized on a single flexible solar array model. A large amount of analytical data are provided at the space station system level, which is conducive to the fine design of each flexible solar array structure. Taking the flexibility of the core module as an example, its displacement nephogram at a certain time is shown in Figure 15. The color indicates the magnitude of the displacement. Among them, red means the largest displacement, blue means the smallest, and the middle color is evenly transitioned. From the displacement nephogram, the absolute displacement distribution and magnitude of each part of the flexible solar array at the time can be seen. Other types of response information such as acceleration and stress at each part of the substructure can also be visualized on a single flexible solar array in a similar way.

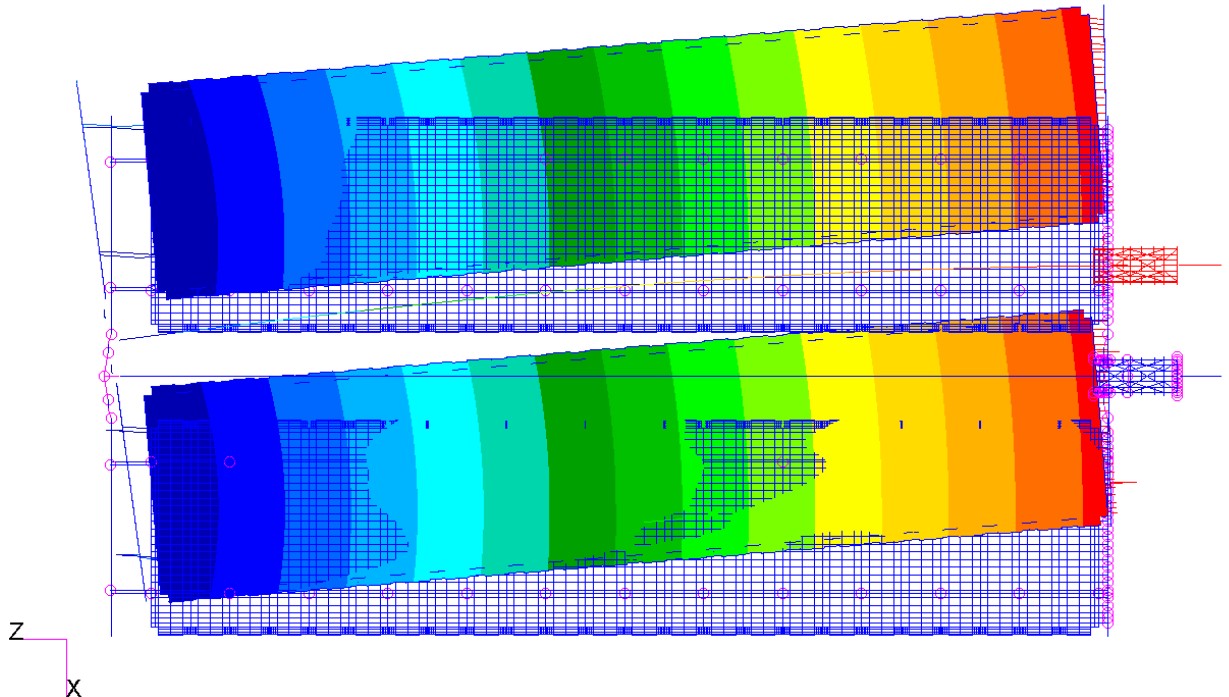

**Figure 15.** The displacement nephogram of the flexible solar array of the core module at a certain time.

### 3.5.2. Acceleration

The acceleration of each part of the flexible solar array is obtained by differentiating the displacement. Figure 16 shows the installation position of the acceleration sensor in the on-orbit test of the space station. In order to facilitate the comparison with the actual on-orbit test results, the time-domain response information of the installation position of the three acceleration sensors was first analyzed and calculated; the result is shown in Figure 17.

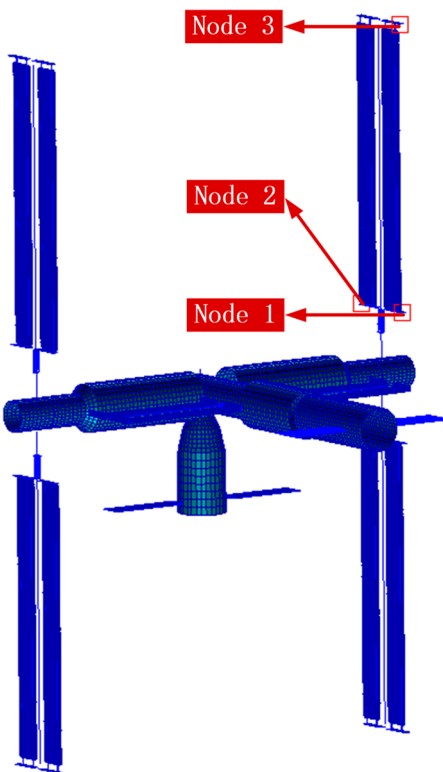

**Figure 16.** Location of acceleration sensor on flexible solar array of Wentian module.

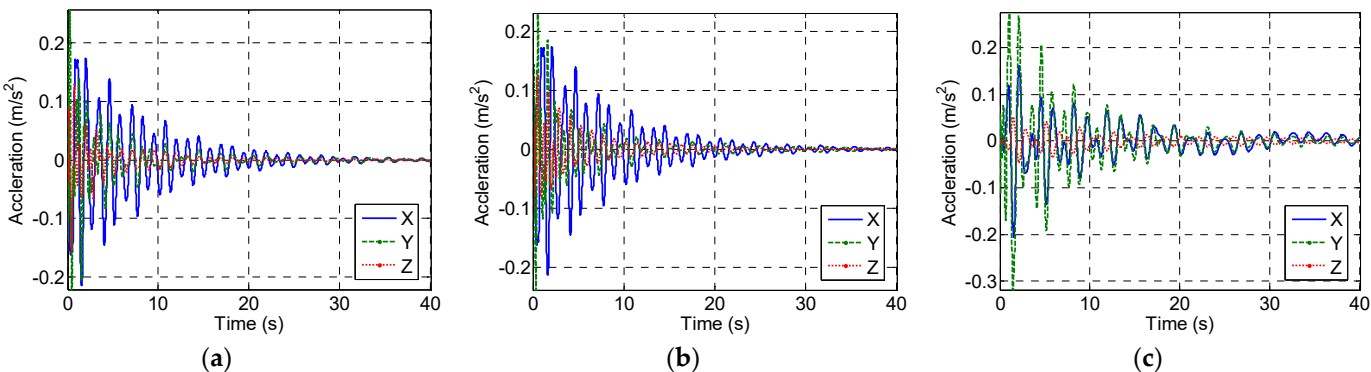

**Figure 17.** Acceleration response of the acceleration sensor installation position. (**a**) Node 1; (**b**) Node 2; (**c**) Node 3.

### 3.5.3. Element Force

The element load is the most concerned in the design of the flexible solar array, especially the root load of the whole array and the extension mechanism, as it is the input source for verifying whether the flexible solar array structure scheme is reasonable and feasible. When the manned spacecraft SZ-15 docked to the space station, the configuration of the space station was basically symmetrical along the Z axis, and the load of the flexible solar arrays of the two core modules was basically the same, as was the load of the four flexible solar arrays of the lab module. Therefore, one flexible solar array in each of the two categories is selected to analyze the load calculation results, shown as Figures 18 and 19, respectively. The plane 1 refers to the out-plane direction of the flexible solar array, and the plane 2 refers to the in-plane direction of the flexible solar array. The description is consistent with the description of the mode modes in Section 3.2.2.

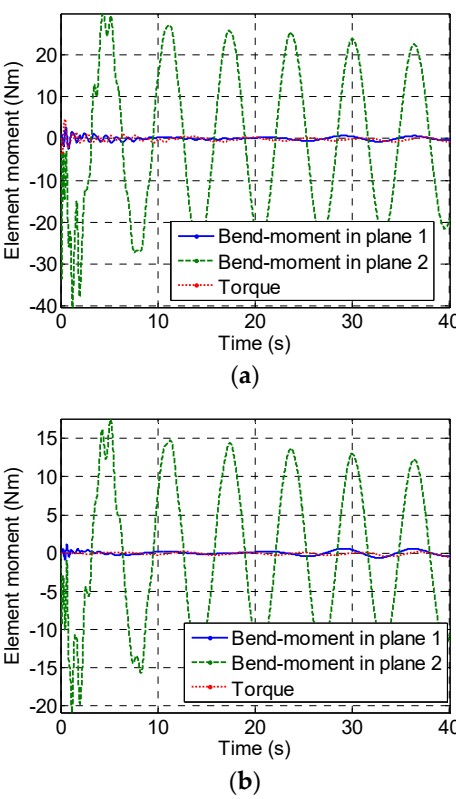

**Figure 18.** Load on the flexible solar array of the core module. (**a**) The root of flexible solar array; (**b**) the root of extension arm.

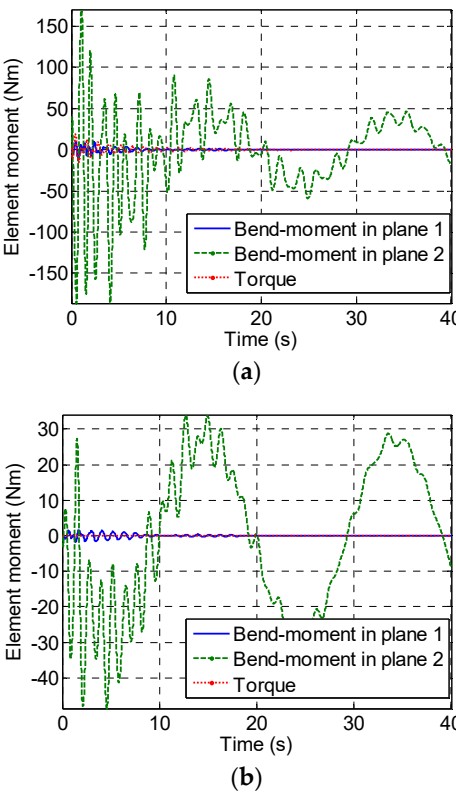

**Figure 19.** Load on the flexible solar array of the lab module. (**a**) The root of flexible solar array; (**b**) the root of extension arm.

## 4. On-Orbit Testing and Verification

### 4.1. On-Orbit Test Scheme

There are many kinds of sensors arranged on the space station to monitor the operation status and the acceleration sensor is one of them, used to measure the vibration acceleration of the solar array. Since no cable can be laid on the solar array, the acceleration sensor used for on-orbit response measurement needs to use dotted WIFI to transmit the measurement information and send the measurement information to the measurement and control subsystem. Therefore, the acceleration sensor needs to have the ability of self-powering and self-charging, and can be used intermittently. The vibration data of the flexible solar array measured using the acceleration sensor have the following uses: (1). Identifying the dynamic parameters of the solar array, such as the modal frequency and damping; (2). The identified solar array dynamics parameters are used to modify the dynamics simulation model, providing a reference for subsequent calculation and analysis; (3). The GNC subsystem can optimize the on-orbit control effect by injecting and modifying the control parameters in the software according to the identified solar array dynamics parameters.

Due to the large flexibility of the flexible solar array and the need to be expanded in-orbit, the acceleration sensor is arranged on the collection box frame of the flexible solar array of the lab module. The location and reference coordinate system are shown in Figure 20. The acceleration sensor code is marked as AP1 (acceleration point 1), AP2 and AP3, respectively. It corresponds to Node 1, Node2, and Node3 as shown in Figure 16. The X direction is along the short edge of the flexible solar array, the Y direction is along the development direction of the extension mechanism, and the Z direction is perpendicular to the flexible solar array.

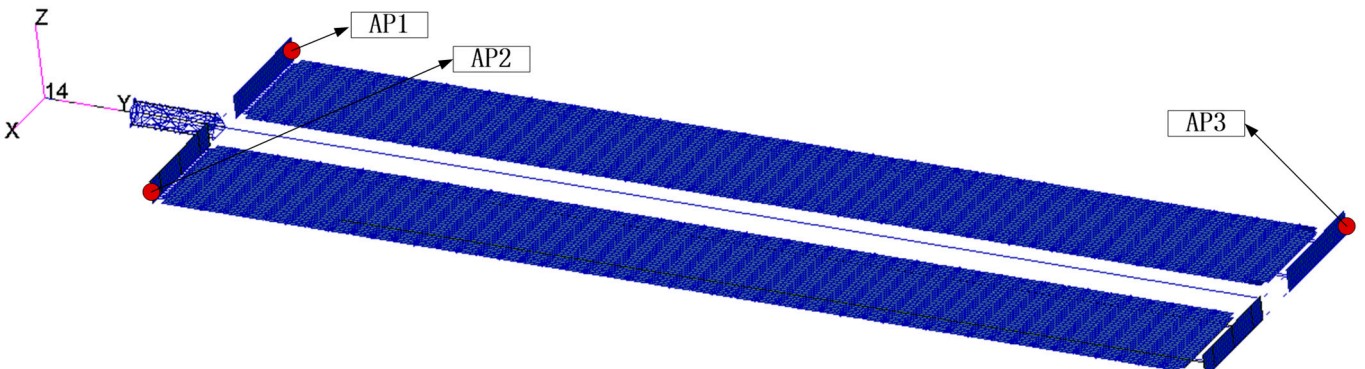

**Figure 20.** The position and reference direction of the acceleration sensor.

The basic composition of the acceleration sensor is shown in Figure 21, which mainly includes energy collection and the power management unit, the energy storage unit, the acceleration body, signal acquisition, the processing and control unit, the wireless communication unit and other components.

In order to improve the identification accuracy, corresponding requirements are put forward for the weight of the sensor, data sampling frequency, on-orbit temperature and other space environment adaptability. Specific technical indicators are shown in Table 9.

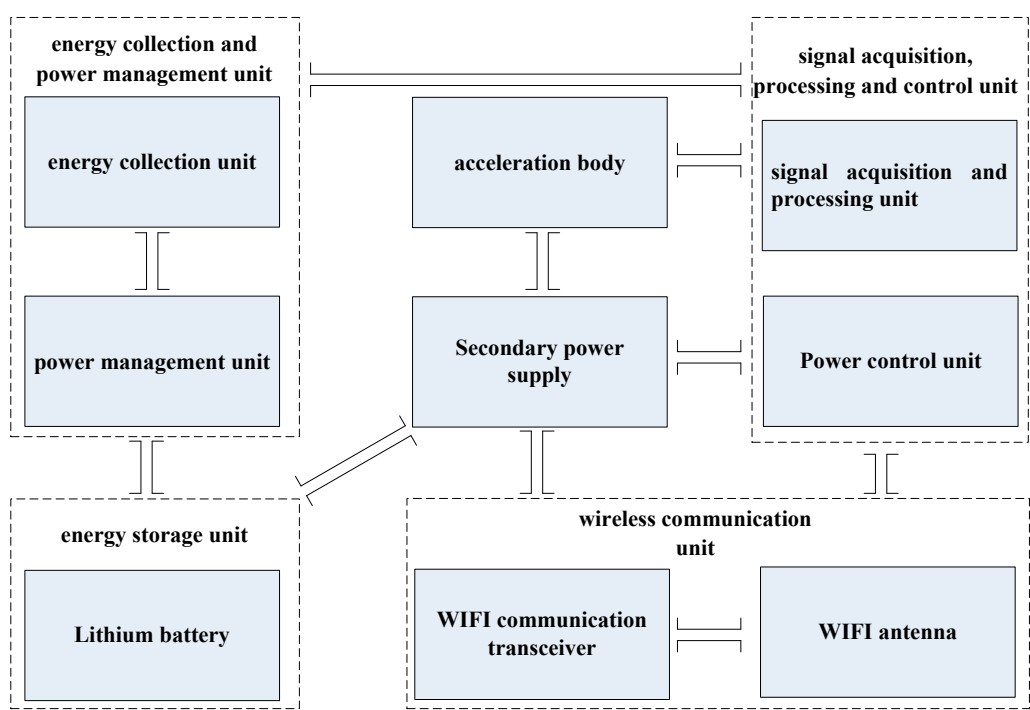

**Figure 21.** The basic composition of the acceleration sensor.

**Table 9.** The main technical index of acceleration sensor.

| No. | Indicator Name | | The Required Range of Indicators |
|---|---|---|---|
| 1. | Sensitivity factor | | $\nleq 10$ V/g |
| 2. | Range of measurement | | (0~1) g |
| 3. | Accuracy of measurement | $\leq 1$ g | $\leq 5 \times 10^{-4}$ g |
| 4. | | $>1$ g | Ensure safety |
| 5. | Zero bias | | $\leq 5 \times 10^{-3}$ g |
| 6. | Dynamic response frequency band | | $>5$ Hz |
| 7. | Data update rate | | 5 Hz |
| 8. | Capacity of lithium battery | | $\nleq 16.0$ Wh |
| 9. | weight | | $\ngeq 1$ kg |
| 10. | Power dissipation | | $\ngeq 5$ W |
| 11. | Dimensions of appearance | | $110 \times 140 \times 100$ (mm) |
| 12. | Maximum single working time | | $\nleq 1$ h |
| 13. | Working temperature | | $-90\,°C{\sim}90\,°C$ |

### 4.2. Results and Validation

Figure 22 is the state of a flexible solar array outside the module during the steady operation of the space station. When SZ-15 docked with the space station, the acceleration sensor was set to work mode to measure the acceleration during docking and transmit the acceleration vibration signal to the ground. Since the reference coordinate system of the acceleration sensor is inconsistent with the analysis coordinate system, in order to facilitate data comparison, the acceleration vibration signal needs to be converted to the analysis coordinate system shown in Figure 4. The comparison between the on-orbit test and analysis results is shown in Figure 23, and the maximum peak results of on-orbit test and analysis are shown in Table 10.

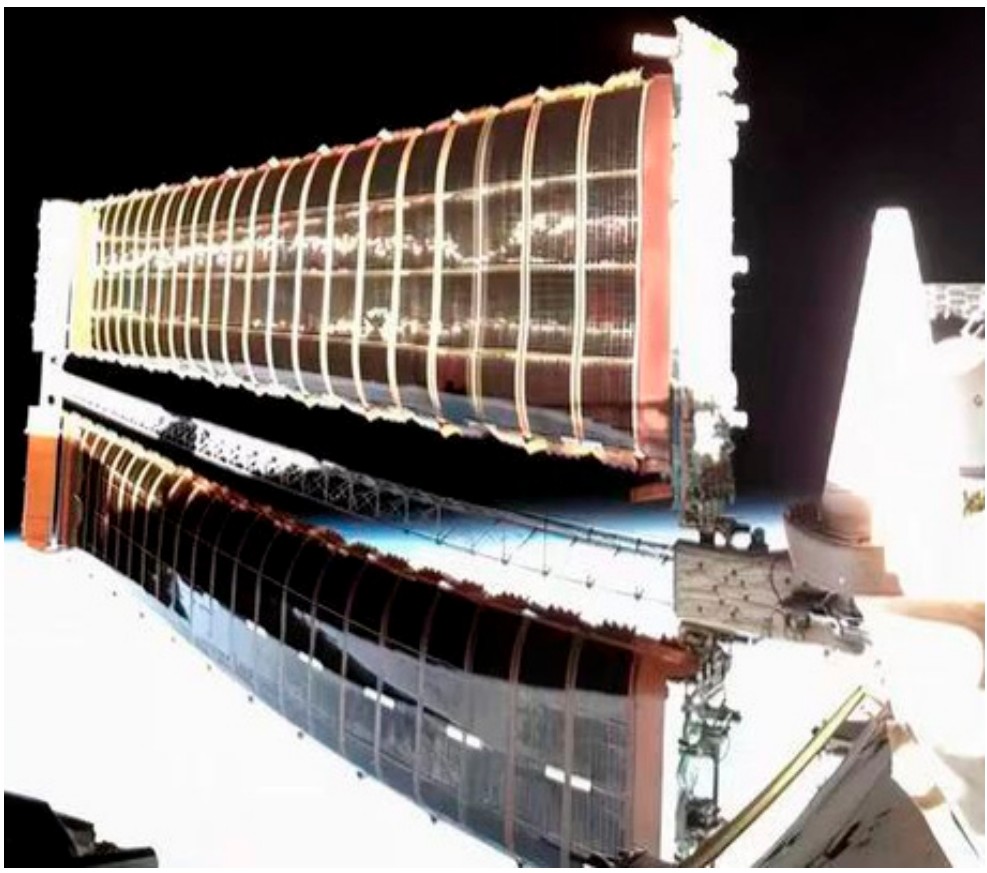

**Figure 22.** On-orbit state of the flexible solar array.

It can be seen from the test results that the magnitude of the measurement data of the acceleration sensor is small, the curve is not smooth due to the small sampling frequency, and the measurement noise signal is more obvious. The error of the acceleration sensor in-orbit test mainly comes from two aspects. First, the test accuracy of the acceleration sensor is 0.005 m/s$^2$ and the background noise signal reaches 0.005 m/s$^2$, as shown in Figure 24 and colors representing the different directions of the acceleration, which is not a small error for the docking process of a small magnitude. Second, the time sampling frequency of the acceleration sensor is 5 Hz, while the typical vibration frequency of the space station assembly when SZ-15 is docked to the space station is about 0.85 Hz (this is the analytical value, and the actual value may be higher), which may lead to missing the maximum peak value in the measurement results. For example, the two measurement points of symmetrical arrangement AP1 and AP2 have a large difference in Y-direction acceleration response.

From Figure 23, the attenuation trend of the acceleration curve in-orbit test and analysis is basically the same, and the average error of the maximum peak acceleration in the docking direction (X direction) is about 20%. The error may also come from the uncertainty of the initial speed of the docking. The median relative speed applied in the analysis is 0.25 m/s, but the actual speed has a deviation of $\pm0.1$ m/s. On the whole, the analysis results of the docking process are in good agreement with the test in general, which has a good guiding significance for engineering design.

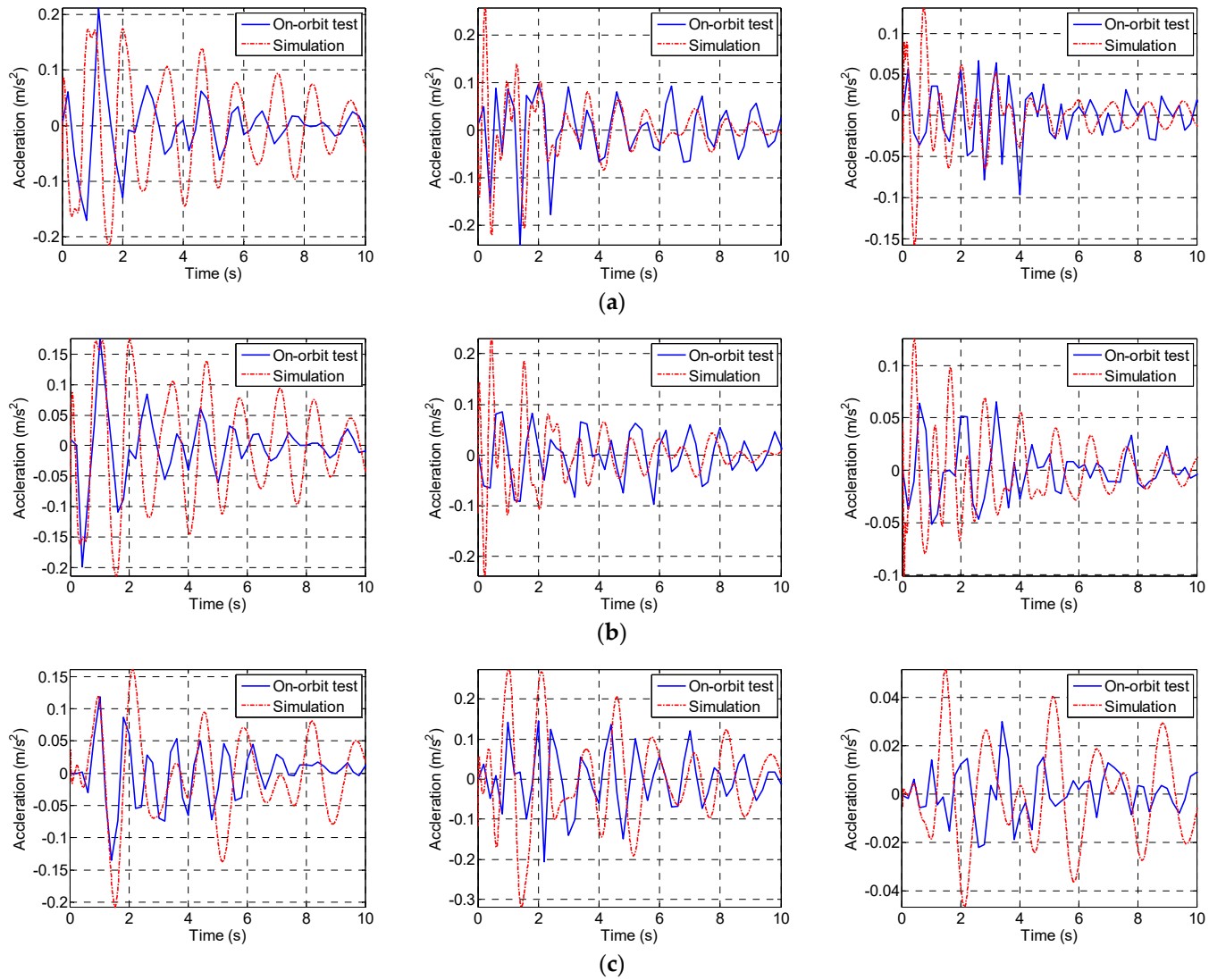

**Figure 23.** The comparison between on-orbit test and analysis results. (**a**) The x, y, z signal of AP1;
(**b**) the x, y, z signal of AP2; (**c**) the x, y, z signal of AP3.

**Table 10.** The maximum peak results of on-orbit test and analysis (unit: m/s$^2$).

| Sensor Code | Orientation | Test | Fem | (Fem-Test)/Test (%) |
|---|---|---|---|---|
| 1. AP1 | 2. X | 0.212 | 0.214 | 0.9% |
| | 3. Y | 0.242 | 0.257 | 6.2% |
| | 4. Z | 0.096 | 0.158 | 64.6% |
| 5. AP2 | 6. X | 0.200 | 0.214 | 7.0% |
| | 7. Y | 0.098 | 0.239 | 143.9% |
| | 8. Z | 0.065 | 0.126 | 93.8% |
| 9. AP3 | 10. X | 0.134 | 0.207 | 54.5% |
| | 11. Y | 0.206 | 0.319 | 54.9% |
| | 12. Z | 0.030 | 0.052 | 73.3% |

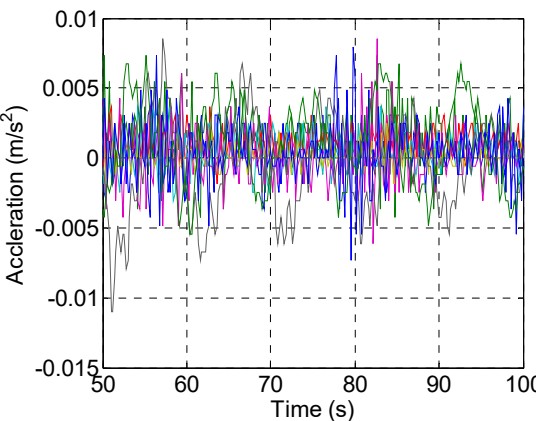

**Figure 24.** Acceleration background noise signal during steady operating.

## 5. Analysis and Discussion

In order to improve the reliability of the flexible solar array, it is necessary to further identify the sensitive factors affecting the on-orbit loads of the flexible solar array in the engineering design. On the one hand, the maximum load it can bear is determined, and on the other hand, the design means are adopted to avoid large loads in the orbit section. First, the initial speed error of $\pm0.1$ m/s is a sensitive factor. In addition, according to the dynamic characteristics of the space station system in Section 3.3, when the SZ-15 manned spacecraft docked and collided with the space station assembly, the free vibration of the residual structure of the space station was aroused, and the synchronous oscillation of the flexible solar arrays was driven. Therefore, the on-orbit load of the flexible solar array is closely related to the coupling degree of the in-plane bending frequency and the X-mode frequency of the space station. In order to analyze the influence degree of different sensitive factors for the on-orbit load of the flexible solar array, the root load of the flexible solar array and the extended structure is selected as the object of comparison.

### 5.1. Impact of Docking Speed Deviation

When SZ-15 docked to the space station, the initial docking speed was controlled within the mean value of 0.25 m/s and the deviation of 0.1 m/s. It is necessary to analyze the impact of the deviation of the initial docking speed. The initial docking speed of SZ-15 was set at 0.15 m/s and 0.35 m/s, respectively, and the same methods and procedures were used to carry out the analysis. Since the in-plane bending moment is the main load, it is compared here. The root loads of the flexible solar array and extension mechanism were calculated, and the maximum values were calculated as shown in Table 11.

**Table 11.** The root loads of the flexible solar array at different docking speeds.

| Case No. | Dock Velocity (Unit: m/s) | Flexible Solar Array of Core Module (Unit: Nm) | | Flexible Solar Array of Lab Module (Unit: Nm) | |
|---|---|---|---|---|---|
| | | The Root of Flexible Solar Array | The Root of Extension Mechanism | The Root of Flexible Solar Array | The Root of Extension Mechanism |
| 1. | 0.15 | 24.18 | 12.54 | 111.79 | 29.17 |
| 2. | 0.25 | 40.29 | 20.90 | 186.32 | 48.62 |
| 3. | 0.35 | 56.41 | 29.26 | 260.85 | 68.07 |

Taking the analysis condition of the docking speed of 0.15 m/s as reference, the analysis results of the docking speed of 0.35 m/s and 0.25 m/s are 1.67 times and 2.33 times of the analysis results of the docking speed of 0.15 m/s, respectively, and shows that the root load of the flexible solar array and the extension mechanism is basically linear with the initial docking velocity of SZ-15.

### 5.2. The Effect of Coupling Frequency

During the docking of SZ-15, the in-plane bending frequency of the flexible solar array and the x-mode frequency of the residual structure of the space station are a pair of coupling frequencies. The resonance of the two frequencies may have a great influence on the result. The frequency of the flexible solar array changes little. Here, we adjust the coupling frequency corresponding to the residual structure, and calculate the root load of the flexible solar array and the extension mechanism. The loads at the root of the flexible solar array and extension mechanism at different coupling frequencies are shown in Table 12.

**Table 12.** The loads at the root of the flexible solar array and extension mechanism at different coupling frequencies.

| Case No. | Frequency of Residual Structure (Unit: Hz) | Flexible Solar Array of Core Module (Unit: Nm) | | Flexible Solar Array of Lab Module (Unit: Nm) | |
|---|---|---|---|---|---|
| | | The Root of Flexible Solar Array | The Root of Extension Mechanism | The Root of Flexible Solar Array | The Root of Extension Mechanism |
| 1. | 0.051 | 37.59 | 20.63 | 135.92 | 94.98 |
| 2. | 0.101 | 49.69 | 27.09 | 59.73 | 49.71 |
| 3. | 0.162 | 93.48 | 51.38 | 50.86 | 40.98 |
| 4. | 0.227 | 33.84 | 20.18 | 75.05 | 33.42 |
| 5. | 0.321 | 31.00 | 18.89 | 88.08 | 29.41 |
| 6. | 0.453 | 31.49 | 17.66 | 182.93 | 44.98 |
| 7. | 0.637 | 36.31 | 18.67 | 165.44 | 62.32 |
| 8. | 0.849 | 40.29 | 20.90 | 186.32 | 48.62 |

In order to intuitively understand the variation trend of the load at the root of the flexible solar array under different coupling frequencies and clearly show the coupling relationship between the load at the root of the flexible solar array and the frequency of the residual structure of the space station, Figures 25 and 26 show the in-plane bending moment of the root of the flexible solar array in the core module and the experimental module under different coupling frequencies. It can be found from the curve that the oscillation frequency of the flexible solar array root load of the core module is mainly its in-plane bending frequency of 0.16 Hz, while the experimental module root load contains relatively rich frequency components, which are closely related to the frequency of the residual structure of the space station. The comparison of Table 12 shows that when the residual structure frequency of the space station is 0.16 Hz, which is the same as the in-plane bending frequency of the flexible solar array of the core module, the root load of the flexible solar array and extension mechanism of the core module increases significantly, while the load of the flexible solar array of the lab module does not change much. When the residual structure frequency of the space station is 0.05 Hz, which is the same as the in-plane bending frequency of the flexible solar array of the lab module, the root load of the flexible solar array and the extension mechanism of the lab module increases significantly, while the load of the flexible solar array of the core module does not change much. This indicates that the coupling degree between the residual structure of the space station and the in-plane bending frequency of the flexible solar array presents a significant positive correlation with the loads of flexible solar array.

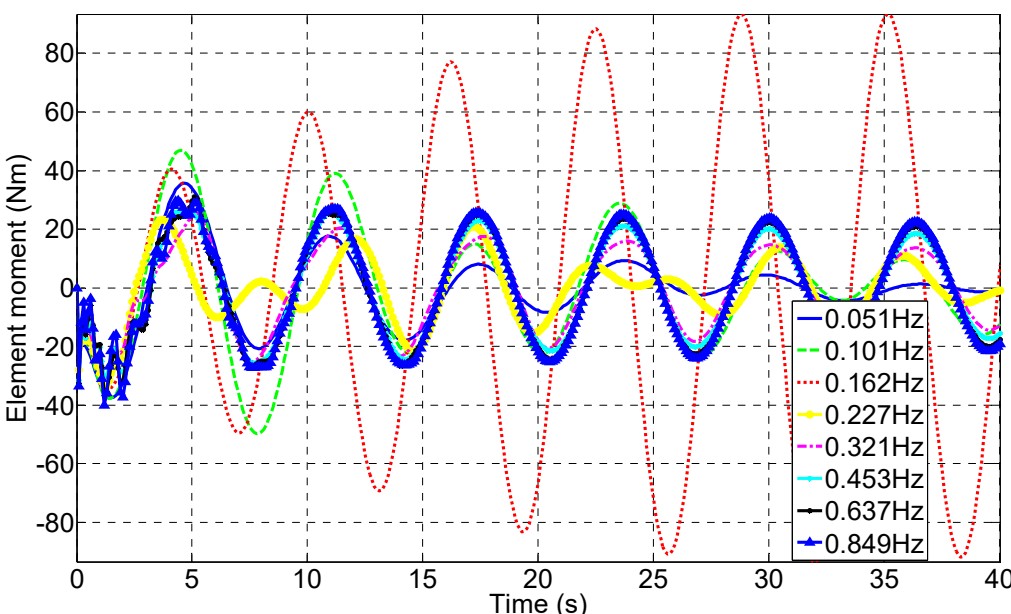

**Figure 25.** In-plane bending moment of the root of flexible solar array of core cabin under different coupling frequency.

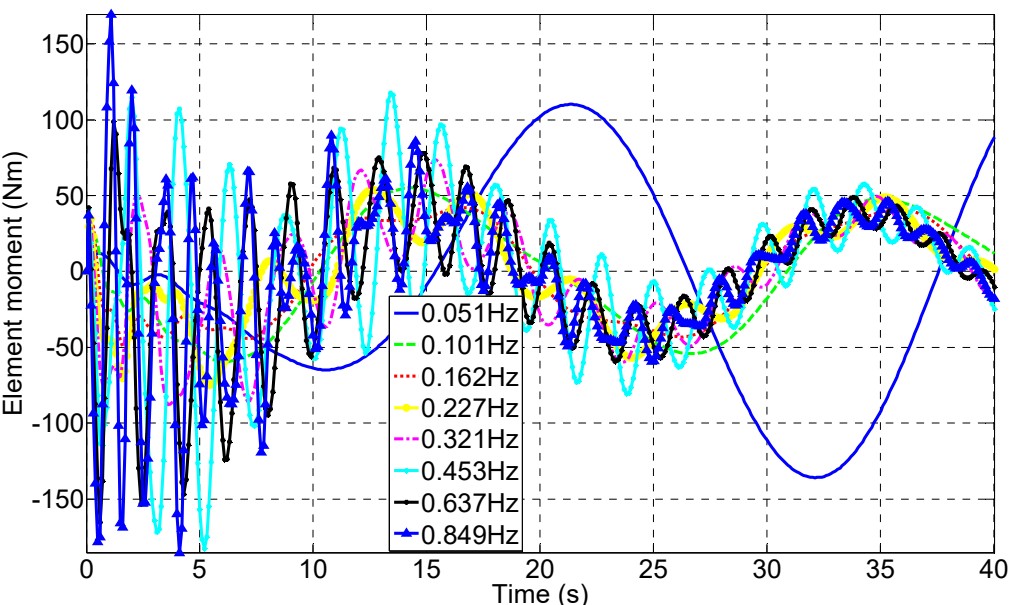

**Figure 26.** In-plane bending moment of the root of flexible solar array of lab module under different coupling frequency.

## 6. Conclusions

To meet the need of the rapid on-orbit coupling dynamics analysis of space assembly containing multiple nonlinear flexible solar arrays in the development of the China Space Station, a nonlinear substructure reduced method considering differential stiffness is proposed, the linearization of nonlinear flexible solar arrays is carried out, and the basic process of internal response data recovery of a reduced model is derived in modal space. Taking the docking of the SZ-15 manned spacecraft to the space station as the analysis object, the coupling dynamics model of the structure of each module and the flexible solar array of the space station was established, and the response information of node displacement and the acceleration of the flexible solar array, as well as the element internal force, etc., were calculated during the docking process. The analysis model and method were tested

and verified according to the measured acceleration in-orbit, and the results were in good agreement. It shows that the modeling and analysis method have better analysis accuracy and can effectively guide the practical engineering design. At the same time, through the iterative analysis of the sensitive factors affecting the on-orbit load of the flexible solar array, it is found that the frequency coupling degree of the space station assembly and the flexible solar array is closely related to the on-orbit load of the flexible solar array. In order to reduce the dynamic response of the flexible solar array in-orbit, the corresponding frequencies of the space station assembly and the flexible solar array need to be designed away from each other during the design stage.

The proposed rapid coupling analysis method for large flexible spacecraft based on reduction and recovery has general versatility and good analytical accuracy. It not only has the advantages of extracting arbitrary position response from the full physical model, but also significantly improves the computational efficiency, which can adapt to the rapid iterative design of engineering, and guide and promote the model development schedule.

**Author Contributions:** Conceptualization, S.W.; Methodology, S.W.; Software, Y.C.; Validation, S.W. and H.Y.; Resources, G.T.; Data curation, H.Y.; Visualization, S.W. and Y.Z.; Funding acquisition, G.T. All authors have read and agreed to the published version of the manuscript.

**Funding:** This research received no external funding.

**Data Availability Statement:** The data presented in this study are available in the article.

**Conflicts of Interest:** The authors declare no conflict of interest.

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
