# Peer review of "A Framework for Rapidly Predicting the Dynamics of Flexible Solar Arrays in the China Space Station with a Verification Based on On-Orbit Measurement Data"

_aerospace, doi:10.3390/aerospace11050411_

Round 1
Reviewer 1 Report
Comments and Suggestions for Authors
The paper is interesting as it deals with a significant issue in space engineering, and offers a comparison of theory and in-orbit data. However, due to the current syntax and clarify of text, and many other issues, I would recommend a major revision. Also the structure is not so good, as some concepts are mentioned, but then only explained later (as the "residual structure" concept). Overall, the context may be acceptable but they way it is presented currently not.
Please find the main aforementioned issues hereafter, but the authors should read and correct the paper thoroughly to have it ready for publication:
- "Facing the current situation that it is difficult to conduct" --> it is not just a current situation, it is a traditional problem for spacecraft equipped with large flexible appendages --> please update the sentence
- "multiple linear and nonlinear flexible structures" what do the authors mean with linear/nonlinear? Is it referred to the proper way of modelling their dynamic behaviour? Is it referred to their geometry shape? It is not clear, please specify
- "It adapts to the fast iterative design requirement of space complex system in engineering practice. " --> not clear, do you refer to the control-structure-material iterative design processes? Please clarify what "adapts" to what.
- " the acceleration of the typical part of the flexible wing" --> which is the typical part of the flexible wing? Please clarify.
- line 139: which type of load is applied to the solar panels?
- line 160: what do you mean by "circular frequency"? you mean natural frequency?
- Par. 2.1 no information is given on transformation between the full set of modal parameters and the condensed one --> which theory is here applied? Please provide more explanation and/or references
- line 165-167: what do you mean by "accessory"? overall the sentence is not clear
- line 178: for fixed interfaces attached to one central residual part also try to see the Imbert's method, which is a semplification of Craig-Bampton. Just as a suggestion, no need to address it in this paper.
- line 249: "Divide" --> change in, for instance, let us divide...
- Fig. 4 --> please mention that you are using a commercial finite element software, which is MSC Nastran judging from the images.
- line 279-281 : not clear at all what you mean here.
- line 334-339 please be more specific about the reason to use linear and non-linear springs and which components are representing.
- line 353: "on the basis" please clarify if you used the results from non-linear analysis as initial pre-stress for the modal analysis?
- line 402 "it can be concluded by the conclusions"... please revise
- line 644: why "linearization", does the model of the solar arrays not include non-linear springs in the FEM model? Why do you need to linearize the model if you are using a commercial FEM model that can perform non-linear analysis as you did in previous sections? Please clarify this.
The authors should revise the references, as some of them seem not coherent with the text, for instance:
- references [9] and [10] should point the reader to the theory of the sub-structuring dynamic modelling, not to papers apparently not related to this: [9] Gong Y, Liu X K, Cao Y, et al. Cosmology from the Chinese Space Station Optical Survey (CSS-OS)[J]. ASTRO- 684 PHYSICAL JOURNAL, 2019,883(2). 685 [10] Wang J X, Li Y Z, Liu X D, et al. Recent active thermal management technologies for the development of ener- 686 gy-optimized aerospace vehicles in China[J]. CHINESE JOURNAL OF AERONAUTICS, 2021,34(2):1-27
Comments on the Quality of English Language
The English syntax should be significantly and extensively revised: often the read cannot understand the meaning of the sentences and those are generally too long and difficult to read. Please revise the manuscript by shortening sentences a and properly re-arranging and re-writing them (for instance, most of the sentences in the abstract are too long). Also, the authors should check for words repetitions in the same sentence, as there are many cases in the paper.
Author Response
We greatly appreciate the reviewer’s encouraging comments and helpful suggestions. We have now further revised the manuscript according to these constructive suggestions carefully. Please see the following replies in detail.

Reviewer 2 Report
Comments and Suggestions for Authors
The article is devoted to important topic of space station with flexible elements dynamic analysis. Despite the good agreement between some analytical results and on-orbit measurements, I do not think that the proposed method is convincing. The entire method and models used in it are described in paragraphs 2.1-2.3, and, honestly, it’s difficult to understand anything from these paragraphs, because everything is described quite general. The method, as it is now described and presented, looks like ordinary oscillation equations and does not represent anything new. I would recommend adding a description in the methodology, what motion models were used, whether it takes into account the angular motion of the space station or only the orbital motion, what is taken into account for orbital motion, etc. For example, angular motion of the main structure would greatly affect the motion of substructures and vise versa. In addition, it is necessary to clarify what coordinates are used in simulation: e.g. angles or quaternions, residual structure centre of mass location or something else, as it would greatly effect equations of motion. What kind of external disturbances were taken into account during the simulation? For example, gravity gradient torque for such large structure might be rather high. How the stiffness matrices and damping coefficients were obtained? Since there are several frequencies higher than 5 Hz, it might be worth looking at smaller time step in simulation - it might increase the accuracy of simulation. And also it would be helpful for readers if authors highlight the novelty of this approach in the text.
Author Response
We greatly appreciate the reviewer’s comments and suggestions. In fact, the method used in the paper is the substructure method, which is a very mature structural dynamics method. And this method indeed involves a set of ordinary oscillation equations. In the authors’ opinion, the main innovation of this paper lies in the successful application of the substructure method, a relatively mature method, to the in orbit dynamic response analysis of the China Space Station composed of multiple cabins and flexible attachments. Furthermore, a nonlinear substructure reduced method considering differential stiffness is proposed, so that the dynamic response of the China Space Station can be quickly obtained to guide the design. In this paper, the transient response of the structure under various loads, such as orbit change and docking, were focused on. The orbital and attitude maneuvers of the space station were not taken into account. Therefore, parameters related to the orbit maneuver and attitude maneuver were not considered, such as angular motion of the main structure, the orbital motion and so on. For more details, please see the point-by-point response.

Reviewer 3 Report
Comments and Suggestions for Authors
Remarks on the paper.
1. The authors write (lines 117-120): "Therefore, in order to solve the on-orbit coupling dynamics problem of nonlinear flexible accessories in complex flexible spacecraft assembly, it is needed to find new methods or improve existing methods." What is the improvement of methods in this paper? Explain in detail.
2. The authors write (lines 232-234): "It is necessary to analyse the magnitude of the load on each part of the flexible solar array during the on-orbit docking process, verifying that the flexible solar array has the capacity to meet the on-orbit task." Explain what task are we talking about? What do you mean?
3. The authors write (lines 254-255): "Based on the actual design status, establish a detailed dynamic analysis model for the space station assembly as shown in the Fig. 3." Tell us more about this model. It is not described in the paper.
4. The authors write (line 281): "The first 10 non-zero modal frequencies are shown in Tab. 2." Explain how these data were obtained?
5. Figures 16 and 20 show the sensor layouts. However, their characteristics are missing. This makes it impossible to judge the accuracy of the measurements achieved.
6. Figure 23 shows the results of comparing the modelling and measurement data. Why are the measurements represented by a continuous curve? How were these measurements approximated? What is their accuracy? It is not possible to make a valid comparison based on Figure 23.
7. The authors write (lines 571-573): "On the whole, the analysis results of the docking process are in good agreement with the test in general, which has a good guiding significance for engineering design. 5" Give quantitative estimates of this convergence. This reasoning is clearly insufficient.
The paper needs a major revision.
Author Response
We greatly appreciate the reviewer’s helpful suggestions. We have now further revised the manuscript according to these constructive suggestions carefully. Please see the following replies in detail.

Round 2
Reviewer 1 Report
Comments and Suggestions for Authors
The authors have addressed the comments. However, the references you deleted should be substituted by proper references related to substructuring.
Comments on the Quality of English Language
The english language should be further revised.
Reviewer 2 Report
Comments and Suggestions for Authors
All my questions were answered
Reviewer 3 Report
Comments and Suggestions for Authors
I believe that the authors have sufficiently refined the article and answered all the questions.